# MixedNUTS: Training-Free Accuracy-Robustness Balance via Nonlinearly Mixed Classifiers

**Yatong Bai**                                                                  *yatong_bai@berkeley.edu*
*University of California, Berkeley*

**Mo Zhou**                                                                          *mzhou32@jhu.edu*
*John Hopkins University*

**Vishal M. Patel**                                                                 *vpatel36@jhu.edu*
*John Hopkins University*

**Somayeh Sojoudi**                                                               *sojoudi@berkeley.edu*
*University of California, Berkeley*

**Reviewed on OpenReview:** *https://openreview.net/forum?id=pyD6cujUmL*

## Abstract

Adversarial robustness often comes at the cost of degraded accuracy, impeding real-life applications of robust classification models. Training-based solutions for better trade-offs are limited by incompatibilities with already-trained high-performance large models, necessitating the exploration of training-free ensemble approaches. Observing that robust models are more confident in correct predictions than in incorrect ones on clean and adversarial data alike, we speculate amplifying this "benign confidence property" can reconcile accuracy and robustness in an ensemble setting. To achieve so, we propose "MixedNUTS", a *training-free* method where the output logits of a robust classifier and a standard non-robust classifier are processed by nonlinear transformations with only three parameters, which are optimized through an efficient algorithm. MixedNUTS then converts the transformed logits into probabilities and mixes them as the overall output. On CIFAR-10, CIFAR-100, and ImageNet datasets, experimental results with custom strong adaptive attacks demonstrate MixedNUTS's vastly improved accuracy and near-SOTA robustness – it boosts CIFAR-100 clean accuracy by 7.86 points, sacrificing merely 0.87 points in robust accuracy.

## 1 Introduction

Neural classifiers are vulnerable to adversarial attacks, producing unexpected predictions when subject to purposefully constructed human-imperceptible input perturbations and hence manifesting severe safety risks (Goodfellow et al., 2015; Madry et al., 2018). Existing methods for robust deep neural networks (Madry et al., 2018; Zhang et al., 2019) often suffer from significant accuracy penalties on clean (unattacked) data (Tsipras et al., 2019; Zhang et al., 2019; Pang et al., 2022). As deep learning continues to form the core of numerous products, trading clean accuracy for robustness is understandably unattractive for real-life users and profit-driven service providers. As a result, despite the continuous development in adversarial robustness research, robust models are rarely deployed and practical services powered by neural networks remain non-robust (Ilyas et al., 2018; Borkar & Chen, 2021).

To bridge the gap between robustness research and applications, researchers have strived to reconcile robustness and accuracy (Balaji et al., 2019; Chen et al., 2021; Raghunathan et al., 2020; Rade & Moosavi-Dezfooli, 2021; Liu & Zhao, 2022; Pang et al., 2022; Cheng et al., 2022), mostly focusing on improving robust training

---

This work was supported by grants from ONR and NSF.

Figure 1: Overview of the proposed MixedNUTS classifier. The nonlinear logit transformation, to be introduced in Section 4, significantly improves the accuracy-robustness balance while only introducing three parameters efficiently optimized with Algorithm 1.

methods. Despite some empirical success, the training-based approach faces inherent challenges. Training robust neural networks from scratch is highly expensive. More importantly, training-based methods suffer from performance bottlenecks. This is because the compatibility between different training schemes is unclear, making it hard to combine multiple advancements. Additionally, it is hard to integrate robust training techniques into rapidly improving large models, often trained or pre-trained with non-classification tasks.

To this end, an alternative training-free direction has emerged, relieving the accuracy-robustness trade-off through an ensemble of a standard (often non-robust) model and a robust model (Bai et al., 2024b;a). This ensemble is referred to as the *mixed classifier*, whereas *base classifiers* refers to its standard and robust components. The mixing approach is mutually compatible with the training-based methods, and hence should be regarded as an add-on. Unlike conventional *homogeneous* ensembling, where all base classifiers share the same goal, the mixed classifier considers *heterogeneous* mixing, with one base classifier specializing in the benign attack-free scenario and the other focusing on adversarial robustness. Thus, the number of base classifiers is naturally fixed as two, in turn maintaining a high inference efficiency.

We observe that many robust base models share a benign confidence property: their correct predictions are much more confident than incorrect ones. Verifying such a property for numerous existing models trained via different methods (Peng et al., 2023; Pang et al., 2022; Wang et al., 2023; Debenedetti et al., 2023; Na, 2020; Gowal et al., 2020; Liu et al., 2023; Singh et al., 2023), we speculate that strengthening this property can improve the mixed classifiers' trade-off even without changing the base classifiers' predicted classes.

Based on this intuition, we propose MixedNUTS (Mixed neUral classifiers with Nonlinear TranSformation), a training-free method that enlarges the robust base classifier confidence difference between correct and incorrect predictions and thereby optimizes the mixed classifier's accuracy-robustness trade-off. MixedNUTS applies nonlinear transformations to the accurate and robust base classifiers' logits before converting them into probabilities used for mixing. We parameterize the transformation with only three coefficients and design an efficient algorithm to optimize them for the best trade-off. Unlike (Bai et al., 2024b), MixedNUTS does not modify base neural network weights or introduce additional components and is for the first time efficiently extendable to larger datasets such as ImageNet. MixedNUTS is compatible with various pre-trained standard and robust models and is agnostic to the base model details such as training method, defense norm ($\ell_\infty$, $\ell_2$, etc.), training data, and model architecture. Therefore, MixedNUTS can take advantage of recent developments in accurate or robust classifiers while being general, lightweight, and convenient.

Our experiments leverage AutoAttack (Croce & Hein, 2020) and strengthened adaptive attacks (details in Appendix B) to confirm the security of the mixed classifier and demonstrate the balanced accuracy and robustness on datasets including CIFAR-10, CIFAR-100, and ImageNet. On CIFAR-100, MixedNUTS improves the clean accuracy by 7.86 percentage points over the state-of-the-art non-mixing robust model while reducing robust accuracy by merely 0.87 points. On ImageNet, MixedNUTS is the first robust model to leverage even larger pre-training datasets such as ImageNet-21k. Furthermore, MixedNUTS allows for inference-time adjustments between clean and adversarial accuracy.

## 2 Background and Related Work

### 2.1 Definitions and Notations

This paper uses $\sigma : \mathbb{R}^c \to (0,1)^c$ to denote the standard Softmax function: for an arbitrary $z \in \mathbb{R}^c$, the $i^{\text{th}}$ entry of $\sigma(z)$ is defined as $\sigma(z)_i := \frac{\exp(z_i)}{\sum_{j=1}^c \exp(z_j)}$, where $z_i$ denotes the $i^{\text{th}}$ entry of $z$. Consider the special case

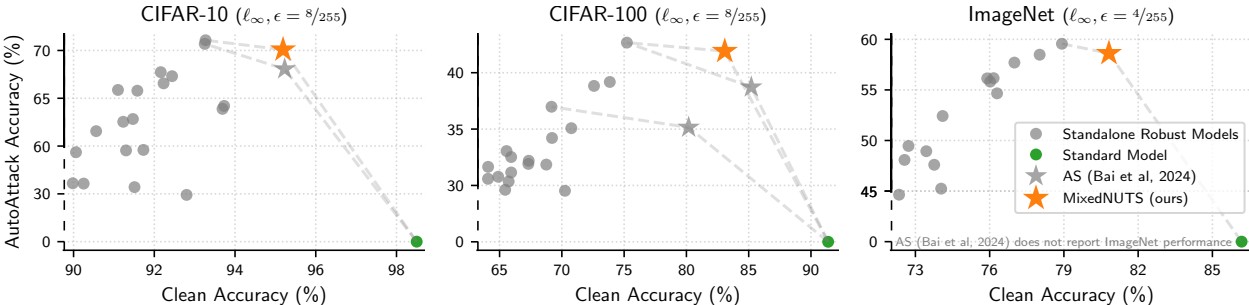

Figure 2: MixedNUTS's accuracy-robustness balance compared to state-of-the-art models on RobustBench. MixedNUTS is more accurate on clean data than all standalone robust models. At the same time, MixedNUTS achieves the second-highest robustness among all models for CIFAR-100 and ImageNet, and is the third most robust for CIFAR-10.

of $z_i = +\infty$ for some $i$, with all other entries of $z$ being less than $+\infty$. We define $\sigma(z)$ for such a $z$ vector to be the basis (one-hot) vector $e_i$. For a classifier $h : \mathbb{R}^d \to \mathbb{R}^c$, we use the composite function $\sigma \circ h : \mathbb{R}^d \to [0, 1]^c$ to denote its output probabilities and use $\sigma \circ h_i$ to denote the $i^{\text{th}}$ entry of it.

We define the notion of *confidence margin* of a classifier as the prediction probability gap between the top two predicted classes:

**Definition 2.1.** Consider a model $h : \mathbb{R}^d \to \mathbb{R}^c$, an arbitrary input $x \in \mathbb{R}^d$, and its associated predicted label $\widehat{y} \in [c]$. The *confidence margin* is defined as

$$m_h(x) := \sigma \circ h_{\widehat{y}}(x) - \max_{i \neq \widehat{y}} \sigma \circ h_i(x).$$

We consider a classifier to be ($\ell_p$-norm) robust at some input $x \in \mathbb{R}^d$ if it assigns the same label to all perturbed inputs $x + \delta$ such that $\|\delta\|_p \leq \epsilon$, where $\epsilon \geq 0$ is the attack radius. We additionally introduce the notion of the worst-case adversarial perturbation in the sense of minimizing the margin:

**Definition 2.2.** Consider an adversarial attack against the confidence margin $m_h(x)$, defined as

$$\min_{\|\delta\| \leq \epsilon} m_h(x + \delta).$$

We define the optimizer of this problem, $\delta_h^\star(x)$, as the *minimum-margin perturbation* of $h(\cdot)$ around $x$. We further define the optimal objective value, denoted as $\underline{m}_h^\star(x)$, as the *minimum margin* of $h(\cdot)$ around $x$.

The attack formulation considered in Definition 2.2 is highly general. Intuitively, when the minimum margin is negative, the adversarial perturbation successfully changes the model prediction. When it is positive, the model is robust at $x$, as perturbations within radius $\epsilon$ cannot change the prediction.

## 2.2 Related Adversarial Attacks and Defenses

While the fast gradient sign method (FGSM) and projected gradient descent (PGD) attacks could attack and evaluate certain models (Madry et al., 2018; Goodfellow et al., 2015), they are insufficient and can fail to attack non-robust models (Carlini & Wagner, 2017; Athalye et al., 2018b; Papernot et al., 2017). To this end, stronger adversaries leveraging novel attack objectives, black-box attacks, and expectation over transformation have been proposed (Gowal et al., 2019; Croce & Hein, 2020; Tramèr et al., 2020). Benchmarks based on these strong attacks, such as RobustBench (Croce et al., 2021), ARES-Bench (Liu et al., 2023), and OODRobustBench (Li et al., 2023), aim to unify defense evaluation.

AutoAttack (Croce & Hein, 2020) is a combination of white-box and black-box attacks (Andriushchenko et al., 2020). It is the attack algorithm of RobustBench (Croce et al., 2021), where AutoAttack-evaluated robust models are often agreed to be trustworthy. We select AutoAttack as the main evaluator, with further strengthening tailored to our defense.

Models aiming to be robust against adversarial attacks often incorporate adversarial training (Madry et al., 2018; Bai et al., 2022; 2023), TRADES (Zhang et al., 2019), or their variations. Later work further enhanced the adversarial robustness by synthetic training data (Wang et al., 2023; Sehwag et al., 2022), data augmentation (Gowal et al., 2020; Rebuffi et al., 2021; Gowal et al., 2021), improved training loss functions (Cui et al., 2023), purposeful architectures (Peng et al., 2023), or efficient optimization (Shafahi et al., 2019). Nevertheless, these methods still suffer from the trade-off between clean and robust accuracy.

To this end, there has been continuous interest from researchers to alleviate this trade-off (Zhang et al., 2019; Lamb et al., 2019; Balaji et al., 2019; Chen et al., 2021; Cheng et al., 2022; Pfrommer et al., 2023; 2024). Most methods are training-based. They are therefore cumbersome to construct and cannot leverage already-trained state-of-the-art robust or non-robust models.

## 2.3 Ensemble and Calibration

Model ensembles, where the outputs of multiple models are combined to produce the overall prediction, have been explored to improve model performance (Ganaie et al., 2022) or estimate model uncertainty (Liu et al., 2019). Ensembling has also been considered to strengthen adversarial robustness (Pang et al., 2019; Adam & Speciel, 2020; Alam et al., 2022; Co et al., 2022). Theoretical robustness analyses of ensemble models indicate that the robust margins, gradient diversity, and runner-up class diversity all contribute to ensemble robustness (Petrov et al., 2023; Yang et al., 2022).

These existing works usually consider *homogeneous* ensemble, meaning that all base classifiers share the same goal (better robustness). They often combine multiple robust models for incremental improvements. In contrast, this work focuses on *heterogeneous* mixing, a fundamentally different paradigm. Here, the base classifiers specialize in different data and the mixed classifier combines their advantages. Hence, we focus on the *two-model* setting, where the role of each model is clearly defined. Specifically, the accurate base classifier specializes in clean data and is usually non-robust, and the robust base classifier excels in adversarial data.

A parallel line of work considers mixing neural model weights (Ilharco et al., 2022; Cai et al., 2023). They generally require all base classifiers to have the same architecture and initialization, which is more restrictive.

Model calibration often involves adjusting confidence, which aims to align a model's confidence with its mispredicting probability, usually via temperature scaling (Guo et al., 2017; Yu et al., 2022; Hinton et al., 2015). While adjusting the confidence of a single model generally does not change its prediction, this is not the case in the ensemble setting. Unlike most calibration research focusing on uncertainty, this paper adjusts the confidence for performance.

## 2.4 Mixing Classifiers for Accuracy-Robust Trade-Off

Consider a classifier $g_{\text{std}} : \mathbb{R}^d \to \mathbb{R}^c$, whose predicted logits are $g_{\text{std},1}, \ldots, g_{\text{std},c}$, where $d$ is the input dimension and $c$ is the number of classes. We assume $g_{\text{std}}(\cdot)$ to be a standard classifier trained for high clean accuracy (and hence may not manifest adversarial robustness). Similarly, we consider another classifier $h_{\text{rob}} : \mathbb{R}^d \to \mathbb{R}^c$ and assume it to be robust against adversarial attacks. We use *accurate base classifier (ABC)* and *robust base classifier (RBC)* to refer to $g_{\text{std}}(\cdot)$ and $h_{\text{rob}}(\cdot)$.

Mixing the outputs of a standard classifier and a robust classifier improves the accuracy-robustness trade-off, and it has been shown that mixing the probabilities is more desirable than mixing the logits from theoretical and empirical perspectives (Bai et al., 2024b;a). Here, we denote the proposed mixed model with $f_{\text{mix}} : \mathbb{R}^d \to \mathbb{R}^c$. Specifically, the $i^{\text{th}}$ output logit of the mixed model follows the formulation

$$f_{\text{mix},i}(x) \coloneqq \log\left((1-\alpha) \cdot \sigma \circ g_{\text{std},i}(x) + \alpha \cdot \sigma \circ h_{\text{rob},i}(x)\right) \tag{1}$$

for all $i \in [c]$, where $\alpha \in [1/2, 1]$ adjusts the mixing weight[1]. The mixing operation is performed in the probability space, and the natural logarithm maps the mixed probability back to the logit space without changing the predicted class for interchangeability with existing models. If the desired output is the probability $\sigma \circ f_{\text{mix}}(\cdot)$, the logarithm can be omitted.

---

[1]Bai et al. (2024b;a) have shown that $\alpha$ should be no smaller than $1/2$ for $f_{\text{mix}}(\cdot)$ to have non-trivial robustness.

# 3 Base Classifier Confidence Modification

We observe that the robust base classifier $h_{\mathrm{rob}}(\cdot)$ often enjoys a benign confidence property: it is much more confident in correct predictions than in mispredictions. I.e., $h_{\mathrm{rob}}(\cdot)$'s confidence margin is much higher when it makes correct predictions. Even if some input is subject to attack (which vastly decreases the confidence margin of correct predictions), if it is correctly predicted, its margin is still expected to be larger than incorrectly predicted natural examples. Section 5.2 verifies this property with multiple model examples, and Appendix C.3 visualizes the confidence margin distributions.

As a result, when mixing the output probabilities $\sigma \circ h_{\mathrm{rob}}(\cdot)$ and $\sigma \circ g_{\mathrm{std}}(\cdot)$ on clean data, where $g_{\mathrm{std}}(\cdot)$ is expected to be more accurate than $h_{\mathrm{rob}}(\cdot)$, $g_{\mathrm{std}}(\cdot)$ can correct $h_{\mathrm{rob}}(\cdot)$'s mistake because $h_{\mathrm{rob}}(\cdot)$ is unconfident. Meanwhile, when the mixed classifier is under attack and $h_{\mathrm{rob}}(\cdot)$ becomes much more reliable than $g_{\mathrm{std}}(\cdot)$, $h_{\mathrm{rob}}(\cdot)$'s high confidence in correct predictions can overcome $g_{\mathrm{std}}(\cdot)$'s misguided outputs. Hence, even when $g_{\mathrm{std}}(\cdot)$'s robust accuracy is near zero, the mixed classifier still inherits most of $h_{\mathrm{rob}}(\cdot)$'s robustness. Combining the above two cases, we can see that the "benign confidence property" of $h_{\mathrm{rob}}(\cdot)$ allows the mixed classifier to simultaneously take advantage of $g_{\mathrm{std}}(\cdot)$'s high clean accuracy and $h_{\mathrm{rob}}(\cdot)$'s adversarial robustness. As a result, modifying and enhancing the base classifiers' confidence has vast potential to further improve the mixed classifier.

Note that this benign confidence property is only observed on robust classifiers. Neural classifiers trained without any robustness considerations often make highly confident mispredictions when subject to adversarial attack. These mispredictions can be even more confident than correctly predicted unperturbed examples, often seeing confidence margins very close to 1. As a result, $g_{\mathrm{std}}(\cdot)$ does not enjoy the benign confidence property, and its confidence property is in general detrimental to the mixture.

## 3.1 Accurate Base Classifier Temperature Scaling

We start with analyzing the accurate base classifier $g_{\mathrm{std}}(\cdot)$, with the goal of mitigating its detrimental confidence property. One approach to achieve this is to scale up $g_{\mathrm{std}}(\cdot)$'s logits before the Softmax operation. To this end, we consider temperature scaling (Hinton et al., 2015). Specifically, we construct the temperature scaled model $g_{\mathrm{std}}^{\mathrm{TS}(T)}(\cdot)$, whose $i^{\mathrm{th}}$ entry is

$$g_{\mathrm{std},i}^{\mathrm{TS}(T)}(x) \coloneqq g_{\mathrm{std},i}(x)/T$$

for all $i$, where $T \geq 0$ is the temperature constant. To scale up the confidence, $T$ should be less than 1.

To understand this operation, observe that temperature scaling increases $g_{\mathrm{std}}(\cdot)$'s confidence in correct clean examples and incorrect adversarial examples simultaneously. However, because $g_{\mathrm{std}}(\cdot)$'s confidence under attack is already close to 1 before scaling, the increase in attacked misprediction confidence is negligible due to the saturation of the Softmax function. Since $g_{\mathrm{std}}(\cdot)$ becomes more confident on correct examples with the mispredicting confidence almost unchanged, its detrimental confidence property is mitigated.

The extreme selection for the temperature $T$ is 0, in which case the predicted probabilities $\sigma \circ g_{\mathrm{std}}^{\mathrm{TS}(0)}(\cdot)$ becomes a one-hot vector corresponding to $g_{\mathrm{std}}(\cdot)$'s predicted class. By scaling with $T = 0$, the detrimental confidence property of $g_{\mathrm{std}}(\cdot)$ is completely eliminated, as a constant margin of precisely 1 is enforced everywhere. Note that we still hope to preserve the ranking of class-wise outputs $g_{\mathrm{std},i}(\cdot)$, so that we can preserve the high accuracy of $g_{\mathrm{std}}(\cdot)$. Given this requirement, eliminating $g_{\mathrm{std}}(\cdot)$'s detrimental confidence property by enforcing a consistent margin is the best one can expect. Appendix D.4 verifies that $T = 0$ produces the best empirical effectiveness among several temperature values. Appendix B.2 discusses how our attacks circumvent the non-differentiability resulting from using $T = 0$.

In addition to eliminating the detrimental confidence property of $g_{\mathrm{std}}(\cdot)$, selecting $T = 0$ also simplifies the analysis on the robust base model $h_{\mathrm{rob}}(\cdot)$ by establishing a direct correlation between $h_{\mathrm{rob}}(\cdot)$'s confidence and the mixed classifier's correctness, thereby allowing for tractable and efficient optimization. Hence, we select $T = 0$ and use $g_{\mathrm{std}}^{\mathrm{TS}(0)}(\cdot)$ as the accurate base classifier for the remaining analyses.

# 4 MixedNUTS – Nonlinearly Mixed Classifier

In contrast to the accurate base classifier, the robust base classifier $h_{\mathrm{rob}}(\cdot)$'s confidence property is benign. To achieve the best accuracy-robustness trade-off with the mixed classifier, we need to *augment* this benign property as much as possible. While a similar temperature scaling operation can achieve some of the desired effects, its potential is limited by applying the same operation to confident and unconfident predictions, and is therefore suboptimal. To this end, we extend confidence modification beyond temperature scaling (which is linear) to allow nonlinear logit transformations. By introducing nonlinearities, we can treat low-confidence and high-confidence examples differently, significantly amplifying $h_{\mathrm{rob}}(\cdot)$'s benign property and thereby considerably enhancing the mixed classifier's accuracy-robustness balance.[2]

## 4.1 Nonlinear Robust Base Classifier Transformation

We aim to build a nonlinearly mapped classifier $h_{\mathrm{rob}}^M(\cdot) := M(h_{\mathrm{rob}}(\cdot))$, where $M \in \mathcal{M} : \mathbb{R}^c \mapsto \mathbb{R}^c$ is a nonlinear transformation applied to the classifier $h_{\mathrm{rob}}(\cdot)$'s logits, and $\mathcal{M}$ is the set of all possible transformations. The prediction probabilities from this transformed robust base model are then mixed with those from $g_{\mathrm{std}}^{\mathrm{TS}(0)}(\cdot)$ to form the mixed classifier $f_{\mathrm{mix}}^M(\cdot)$ following (1). For the optimal accuracy-robustness trade-off, we select an $M$ that maximizes the clean accuracy of $f_{\mathrm{mix}}^M(\cdot)$ while maintaining the desired robust accuracy. Formally, this goal is described as the optimization problem

$$\max_{M \in \mathcal{M}, \ \alpha \in [1/2, 1]} \mathbb{P}_{(X,Y) \sim \mathcal{D}}\Big[\arg\max_i f_{\mathrm{mix},i}^M(X) = Y\Big] \tag{2}$$
$$\text{subject to} \quad \mathbb{P}_{(X,Y) \sim \mathcal{D}}\Big[\arg\max_i f_{\mathrm{mix},i}^M(X + \delta_{f_{\mathrm{mix}}^M}^\star(X)) = Y\Big] \geq r_{f_{\mathrm{mix}}^M},$$

where $\mathcal{D}$ is the distribution of data-label pairs, $r_{f_{\mathrm{mix}}^M}$ is the desired robust accuracy of $f_{\mathrm{mix}}^M(\cdot)$, and $\delta_{f_{\mathrm{mix}}^M}^\star(x)$ is the minimum-margin perturbation of $f_{\mathrm{mix}}^M(\cdot)$ at $x$. Note that $f_{\mathrm{mix},i}^M(\cdot)$ implicitly depends on $M$ and $\alpha$.

The problem (2) depends on the robustness behavior of the mixed classifier, which is expensive to probe. Ideally, the optimization should only need the base classifier properties, which can be evaluated beforehand. To allow such a simplification, we make the following two assumptions.

**Assumption 4.1.** On unattacked clean data, if $h_{\mathrm{rob}}^M(\cdot)$ makes a correct prediction, then $g_{\mathrm{std}}(\cdot)$ is also correct.

Assumption 4.1 allows us to focus on examples correctly classified by the accurate base classifier $g_{\mathrm{std}}^{\mathrm{TS}(0)}(\cdot)$ but not by the robust base model $h_{\mathrm{rob}}^M(\cdot)$ when optimizing the transformation $M(\cdot)$ to maximize the clean accuracy of the mixed classifier. Under Assumption 4.1, we can safely discard the opposite case of $g_{\mathrm{std}}^{\mathrm{TS}(0)}(\cdot)$ being incorrect while $h_{\mathrm{rob}}^M(\cdot)$ being correct on clean data. Assumption 4.1 makes sense because $g_{\mathrm{std}}^{\mathrm{TS}(0)}(\cdot)$'s clean accuracy should be considerably higher than $h_{\mathrm{rob}}^M(\cdot)$'s to justify mixing them together, and training standard classifiers that noticeably outperforms robust models on clean data is usually possible in practice.

**Assumption 4.2.** The transformation $M(\cdot)$ does not change the predicted class due to, *e.g.*, monotonicity. Namely, it holds that $\arg\max_i M(h_{\mathrm{rob}}(x))_i = \arg\max_i h_{\mathrm{rob},i}(x)$ for all $x$.

We make this assumption because we want the logit transformation to preserve the accuracy of $h_{\mathrm{rob}}(\cdot)$. While it is mathematically possible to obtain an $M(\cdot)$ that further increases the accuracy of $h_{\mathrm{rob}}(\cdot)$, finding it could be as hard as training a new improved robust model. Hence, the best one would expect from a relatively simple nonlinear transformation is to enhance $h_{\mathrm{rob}}(\cdot)$'s benign confidence margin property while not changing the predicted class. Later in this paper, we will propose Algorithm 1 to find such a transformation.

These two assumptions allow us to decouple the optimization of $M(\cdot)$ from the accurate base classifier $g_{\mathrm{std}}(\cdot)$. This is because as proven in (Bai et al., 2024b, Lemma 1), the mixed classifier is guaranteed to be robust when $h_{\mathrm{rob}}^M$ is robust with margin no smaller than $\frac{1-\alpha}{\alpha}$ (with the implicit assumption $\alpha \geq 0.5$. Hence, we can solve the following problem as a surrogate for our goal formulation (2):

$$\min_{M \in \mathcal{M}, \ \alpha \in [1/2, 1]} \mathbb{P}_{X \sim \mathcal{X}_{\mathrm{clean}}^{\checkmark}}\Big[m_{h_{\mathrm{rob}}^M}(X) \geq \tfrac{1-\alpha}{\alpha}\Big] \qquad \text{subject to} \qquad \mathbb{P}_{Z \sim \mathcal{X}_{\mathrm{adv}}^{\checkmark}}\Big[m_{h_{\mathrm{rob}}^M}^\star(Z) \geq \tfrac{1-\alpha}{\alpha}\Big] \geq \beta, \tag{3}$$

---

[2]The same nonlinear logit transformation is not applied to the accurate base classifier because its confidence property is *not benign*. As explained in Section 3.1, eliminating $g_{\mathrm{std}}(\cdot)$'s detrimental confidence property by enforcing a constant margin with one-hot encoding is the best one can expect.

where $\mathcal{X}_{\text{clean}}^{\text{✗}}$ is the distribution formed by clean examples incorrectly classified by $h_{\text{rob}}^M(\cdot)$, $\mathcal{X}_{\text{adv}}^{\text{✓}}$ is the distribution formed by attacked examples correctly classified by $h_{\text{rob}}^M(\cdot)$, $X$, $Z$ are the random variables drawn from these distributions, and $\beta \in [0, 1]$ controls the mixed classifier's desired level of robust accuracy with respect to the robust accuracy of $h_{\text{rob}}(\cdot)$.

Note that (3) no longer depends on $g_{\text{std}}(\cdot)$, allowing for replacing the standard base classifier without re-solving for a new transformation $M(\cdot)$. The following two theorems justify approximating (2) with (3) by characterizing the optimizers of (3):

**Theorem 4.3.** *Suppose that Assumption* 4.2 *holds. Let* $r_{f_{mix}^M}$ *and* $r_{h_{rob}}$ *denote the robust accuracy of* $f_{mix}^M(\cdot)$ *and* $h_{rob}(\cdot)$ *respectively. If* $\beta \geq r_{f_{mix}^M}/r_{h_{rob}}$, *then a solution to* (3) *is feasible for* (2).

**Theorem 4.4.** *Suppose that Assumption* 4.1 *holds. Furthermore, consider an input random variable* $X$ *and suppose that the margin of* $h_{rob}^M(X)$ *is independent of whether* $g_{std}(X)$ *is correct. Then, minimizing the objective of* (3) *is equivalent to maximizing the objective of* (2).

The proofs of Theorem 4.3 and Theorem 4.4 are provided in Appendices A.1 and A.2, respectively. In Appendix D.8.1 and Appendix D.8.2, we discuss the minor effects of slight violations to Assumption 4.1 and Assumption 4.2, respectively. Moreover, the independence assumption in Theorem 4.4 can be relaxed with minor changes to our method, which we discuss in Appendix D.8.3. Also note that Theorems 4.3 and 4.4 rely on using $T = 0$ for $g_{\text{std}}(\cdot)$'s temperature scaling, justifying this temperature setting selected in Section 3.1.

## 4.2 Parameterizing the Transformation $M$

Optimizing the nonlinear transformation $M(\cdot)$ requires representing it with parameters. To avoid introducing additional training requirements or vulnerable backdoors, the parameterization should be simple (*i.e.*, not introducing yet another neural network). Thus, we introduce a manually designed transformation with only three parameters, along with an algorithm to efficiently optimize the three parameters.

Unlike linear scaling and the Softmax operation, which are shift-agnostic (*i.e.*, adding a constant to all logits does not change the predicted probabilities), the desired nonlinear transformations' behavior heavily depends on the numerical range of the logits. Thus, to make the nonlinear transformation controllable and interpretable, we pre-process the logits by applying layer normalization (LN): for each input $x$, we standardize the logits $h_{\text{rob}}(x)$ to have zero mean and identity variance. We observe that LN itself also slightly increases the margin difference between correct and incorrect examples, favoring our overall formulation as shown in Figure 6. This phenomenon is further explained in Appendix D.7.

Among the post-LN logits, only those associated with confidently predicted classes can be large positive values. To take advantage of this property, we use a clamping function $\text{Clamp}(\cdot)$, such as ReLU, GELU, ELU, or SoftPlus, to bring the logits smaller than a threshold toward zero. This clamping operation can further suppress the confidence of small-margin predictions while preserving large-margin predictions. Since correct examples often enjoy larger margins, the clamping function enlarges the margin gap between correct and incorrect examples. We provide an ablation study over candidate clamping functions in Appendix D.2 and empirically select GELU for our experiments.

Finally, since the power functions with greater-than-one exponents diminish smaller inputs while amplifying larger ones, we exponentiate the clamping function outputs to a constant power and preserve the sign. Putting everything together, with the introduction of three scalars $s$, $p$, and $c$ to parameterize $M(\cdot)$, the combined nonlinearly transformed robust base classifier $h_{\text{rob}}^{M(s,p,c)}(\cdot)$ becomes

$$h_{\text{rob}}^{M(s,p,c)}(x) = s \cdot \left| h_{\text{rob}}^{\text{Clamp}(c)}(x) \right|^p \cdot \text{sgn}\left( h_{\text{rob}}^{\text{Clamp}(c)}(x) \right) \quad \text{where} \quad h_{\text{rob}}^{\text{Clamp}(c)}(x) = \text{Clamp}\left( \text{LN}(h_{\text{rob}}(x)) + c \right). \quad (4)$$

Here, $s \in (0, +\infty)$ is a scaling constant, $p \in (0, +\infty)$ is an exponent constant, and $c \in \mathbb{R}$ is a bias constant that adjusts the cutoff location of the clamping function. With a slight abuse of notation, $M(s, p, c)(\cdot)$ denotes the transformation parameterized with $s$, $p$, and $c$. In (4), we apply the absolute value before the exponentiation to maintain compatibility with non-integer $p$ values and use the sign function to preserve the sign. Note that when the clamping function is linear and $p = 1$, (4) degenerates to temperature scaling with LN. Hence, an optimal combination of $s$, $p$, and $c$ is guaranteed to be no worse than temperature scaling.

Note that the nonlinear transformation $M(s, p, c)(\cdot)$ generally adheres to Assumption 4.2. While Assumption 4.2 may be slightly violated if GELU is chosen as the clamping function due to its portion around zero being not monotonic, its effect is empirically very small according to our observation, partly because the negative slope is very shallow. We additionally note that the certified robustness results presented in (Bai et al., 2024b) also apply to the nonlinearly mixed classifiers in this work.

With the accurate base classifier's temperature scaling and the robust base classifier's nonlinear logit transformation in place, the overall formulation of MixedNUTS becomes

$$f_{\mathrm{mix}}^{M(s,p,c)}(x) := \log\big((1 - \alpha) \cdot g_{\mathrm{std}}^{\mathrm{TS}(0)}(x) + \alpha \cdot h_{\mathrm{rob}}^{M(s,p,c)}(x)\big), \tag{5}$$

as illustrated in Figure 1.

### 4.3 Efficient Algorithm for Optimizing $s$, $p$, $c$, and $\alpha$

With the nonlinear transformation parameterization in place, the functional-space optimization problem (3) reduces to the following algebraic optimization formulation:

$$\begin{aligned}
&\min_{s,p,c,\alpha \in \mathbb{R}} && \mathbb{P}_{X \sim \mathcal{X}_{\mathrm{clean}}^{\boldsymbol{\chi}}}\big[m_{h_{\mathrm{rob}}^{M(s,p,c)}}(X) \geq \tfrac{1-\alpha}{\alpha}\big] \\
&\text{subject to} && \mathbb{P}_{Z \sim \mathcal{X}_{\mathrm{adv}}^{\checkmark}}\big[\underline{m}_{h_{\mathrm{rob}}^{M(s,p,c)}}^{\star}(Z) \geq \tfrac{1-\alpha}{\alpha}\big] \geq \beta, \quad s \geq 0, \quad p \geq 0, \quad \tfrac{1}{2} \leq \alpha \leq 1.
\end{aligned} \tag{6}$$

Exactly solving (6) involves evaluating $\underline{m}_{h_{\mathrm{rob}}^{M(s,p,c)}}^{\star}(x)$ for every $x$ in the support of the distribution of correctly predicted adversarial examples $\mathcal{X}_{\mathrm{adv}}^{\checkmark}$. This is intractable because the support is a continuous set and the distributions $\mathcal{X}_{\mathrm{clean}}^{\boldsymbol{\chi}}$ and $\mathcal{X}_{\mathrm{adv}}^{\checkmark}$ implicitly depend on the optimization variables $s$, $p$, and $c$. To this end, we approximate $\mathcal{X}_{\mathrm{clean}}^{\boldsymbol{\chi}}$ and $\mathcal{X}_{\mathrm{adv}}^{\checkmark}$ with a small set of data. Consider the subset of clean examples incorrectly classified by $h_{\mathrm{rob}}^{\mathrm{LN}}(\cdot)$, denoted as $\widetilde{\mathcal{X}}_{\mathrm{clean}}^{\boldsymbol{\chi}}$, and the subset of attacked examples correctly classified by $h_{\mathrm{rob}}^{\mathrm{LN}}(\cdot)$, denoted as $\widetilde{\mathcal{X}}_{\mathrm{adv}}^{\checkmark}$. Because we use $h_{\mathrm{rob}}^{\mathrm{LN}}(\cdot)$ instead of $h_{\mathrm{rob}}^{M(s,p,c)}(\cdot)$ to obtain $\widetilde{\mathcal{X}}_{\mathrm{clean}}^{\boldsymbol{\chi}}$ and $\widetilde{\mathcal{X}}_{\mathrm{adv}}^{\checkmark}$, using them as surrogates to $\mathcal{X}_{\mathrm{clean}}^{\boldsymbol{\chi}}$ and $\mathcal{X}_{\mathrm{adv}}^{\checkmark}$ decouples the probability measures from the optimization variables.

Despite optimizing $s$, $p$, $c$, and $\alpha$ on a small set of data, overfitting is unlikely since there are only four parameters, and thus a small number of images should suffice. Appendix D.3 analyzes the effect of the data subset size on optimization quality and confirms the absence of overfitting.

The minimum margin $\underline{m}_{h_{\mathrm{rob}}^{M(s,p,c)}}^{\star}(x)$ also depends on the optimization variables $s$, $p$, $c$, and $\alpha$, as its calculation requires the minimum-margin perturbation for $h_{\mathrm{rob}}^{M(s,p,c)}(\cdot)$ around $x$. Since finding $\underline{m}_{h_{\mathrm{rob}}^{M(s,p,c)}}^{\star}(x)$ for all $s$, $p$, and $c$ combinations is intractable, we seek to use an approximation that does not depend on $s$, $p$, and $c$. Specifically, the approximation is $\widetilde{\underline{m}}_{h_{\mathrm{rob}}^{M(s,p,c)}}(x)$, defined as

$$\widetilde{\underline{m}}_{h_{\mathrm{rob}}^{M(s,p,c)}}(x) := m_{h_{\mathrm{rob}}^{M(s,p,c)}}\big(x + \widetilde{\delta}_{h_{\mathrm{rob}}^{\mathrm{LN}}}(x)\big) \approx m_{h_{\mathrm{rob}}^{M(s,p,c)}}\big(x + \delta_{h_{\mathrm{rob}}^{M(s,p,c)}}^{\star}(x)\big) = \underline{m}_{h_{\mathrm{rob}}^{M(s,p,c)}}^{\star}(x),$$

where $\widetilde{\delta}_{h_{\mathrm{rob}}^{\mathrm{LN}}}(x)$ is an empirical minimum-margin perturbation of $h_{\mathrm{rob}}^{\mathrm{LN}}(\cdot)$ around $x$ obtained from a strong adversarial attack. Note that calculating $\widetilde{\underline{m}}_{h_{\mathrm{rob}}^{M(s,p,c)}}(x)$ does not require attacking $h_{\mathrm{rob}}^{M(s,p,c)}(\cdot)$ and instead attacks $h_{\mathrm{rob}}^{\mathrm{LN}}(\cdot)$, which is independent of the optimization variables, ensuring optimization efficiency. To obtain $\widetilde{\underline{m}}_{h_{\mathrm{rob}}^{M(s,p,c)}}(x)$, in Appendix B.1, we propose *minimum-margin AutoAttack (MMAA)*, an AutoAttack variant that keeps track of the minimum margin while generating perturbations. While some components of MMAA require $h_{\mathrm{rob}}^{\mathrm{LN}}(\cdot)$'s gradient information, Algorithm 1 can still apply after some modifications even if the base classifiers are black boxes with unavailable gradients, with the details discussed in Appendix D.5.

Since the probability measures and the perturbations are now both decoupled from $s$, $p$, $c$, $\alpha$, we only need to run MMAA once to estimate the worst-case perturbation, making this hyper-parameter search problem efficiently solvable. While using $h_{\mathrm{rob}}^{\mathrm{LN}}(\cdot)$ as a surrogate to $h_{\mathrm{rob}}^{M(s,p,c)}(\cdot)$ introduces a distribution mismatch, we expect this mismatch to be benign. To understand this, observe that the nonlinear logit transformation (4) generally preserves the predicted class due to the (partially) monotonic characteristics of GELU and the sign-preserving power function. Consequently, we expect the accuracy and minimum-margin perturbations of $h_{\mathrm{rob}}^{M(s,p,c)}(\cdot)$ to be very similar to those of $h_{\mathrm{rob}}^{\mathrm{LN}}(\cdot)$. Appendix D.9 empirically verifies this speculated proximity.

---

**Algorithm 1** Algorithm for optimizing $s$, $p$, $c$, and $\alpha$.

---

1: Given an image set, save the predicted logits associated with mispredicted clean images $\left\{h_{\text{rob}}^{\text{LN}}(x) : x \in \widetilde{\mathcal{X}}_{\text{clean}}^{\text{✗}}\right\}$.

2: Run MMAA on $h_{\text{rob}}^{\text{LN}}(\cdot)$ and save the logits of correctly classified perturbed inputs $\left\{h_{\text{rob}}^{\text{LN}}(x) : x \in \widetilde{\mathcal{A}}_{\text{adv}}^{\text{✓}}\right\}$.

3: Initialize candidate values $s_1, \ldots, s_l$, $p_1, \ldots, p_m$, $c_1, \ldots, c_n$.

4: **for** $s_i$ for $i = 1, \ldots, l$ **do**

5:     **for** $p_j$ for $j = 1, \ldots, m$ **do**

6:         **for** $c_k$ for $k = 1, \ldots, n$ **do**

7:             Obtain mapped logits $\left\{h_{\text{rob}}^{M(s_i, p_j, c_k)}(x) : x \in \widetilde{\mathcal{A}}_{\text{adv}}^{\text{✓}}\right\}$.

8:             Calculate the margins from the mapped logits $\left\{m_{h_{\text{rob}}^{M(s_i, p_j, c_k)}}(x) : x \in \widetilde{\mathcal{A}}_{\text{adv}}^{\text{✓}}\right\}$.

9:             Store the bottom $1 - \beta$-quantile of the margins as $q_{1-\beta}^{ijk}$ (corresponds to $\frac{1-\alpha}{\alpha}$ in (7)).

10:            Record the current objective $o^{ijk} \leftarrow \mathbb{P}_{X \in \widetilde{\mathcal{X}}_{\text{clean}}^{\text{✗}}}\left[m_{h_{\text{rob}}^{M(s_i, p_j, c_k)}}(X) \geq q_{1-\beta}^{ijk}\right]$.

11:         **end for**

12:     **end for**

13: **end for**

14: Find optimal indices $(i^\star, j^\star, k^\star) = \arg\min_{i,j,k} o^{ijk}$.

15: Recover optimal mixing weight $\alpha^\star := 1/\left(1 + q_{1-\beta}^{i^\star j^\star k^\star}\right)$.

16: **return** $s^\star := s_{i^\star}$, $p^\star := p_{j^\star}$, $c^\star := c_{k^\star}$, $\alpha^\star$.

---

To simplify notations, let $\widetilde{\mathcal{A}}_{\text{adv}}^{\text{✓}} := \left\{x + \widetilde{\delta}_{h_{\text{rob}}^{\text{LN}}}(x) : \ x \in \widetilde{\mathcal{X}}_{\text{adv}}^{\text{✓}}\right\}$ denote all correctly predicted minimum-margin perturbed images for $h_{\text{rob}}^{\text{LN}}(\cdot)$. Inherently, it holds that

$$\mathbb{P}_{Z \in \widetilde{\mathcal{A}}_{\text{adv}}^{\text{✓}}}\left[m_{h_{\text{rob}}^{M(s,p,c)}}(Z) \geq \tfrac{1-\alpha}{\alpha}\right] = \mathbb{P}_{Z \in \widetilde{\mathcal{X}}_{\text{adv}}^{\text{✓}}}\left[\widetilde{m}_{h_{\text{rob}}^{M(s,p,c)}}(Z) \geq \tfrac{1-\alpha}{\alpha}\right] \approx \mathbb{P}_{Z \sim \mathcal{X}_{\text{adv}}^{\text{✓}}}\left[m^\star_{h_{\text{rob}}^{M(s,p,c)}}(Z) \geq \tfrac{1-\alpha}{\alpha}\right].$$

The approximate hyper-parameter selection problem, which can be solved in surrogate to (6), is then

$$
\begin{aligned}
\min_{s,p,c,\alpha \in \mathbb{R}} \quad & \mathbb{P}_{X \in \widetilde{\mathcal{X}}_{\text{clean}}^{\text{✗}}}\left[m_{h_{\text{rob}}^{M(s,p,c)}}(X) \geq \tfrac{1-\alpha}{\alpha}\right] \\
\text{subject to} \quad & \mathbb{P}_{Z \in \widetilde{\mathcal{A}}_{\text{adv}}^{\text{✓}}}\left[m_{h_{\text{rob}}^{M(s,p,c)}}(Z) \geq \tfrac{1-\alpha}{\alpha}\right] \geq \beta, \quad s \geq 0, \quad p \geq 0, \quad 1/2 \leq \alpha \leq 1.
\end{aligned}
\tag{7}
$$

Since (7) only has four optimization variables, it can be solved via a grid search algorithm. Furthermore, the constraint $\mathbb{P}_{Z \in \widetilde{\mathcal{A}}_{\text{adv}}^{\text{✓}}}\left[m_{h_{\text{rob}}^{M(s,p,c)}}(Z) \geq \tfrac{1-\alpha}{\alpha}\right] \geq \beta$ should always be active at optimality.[3] Hence, we can treat this constraint as equality, reducing the searching grid dimension to three. Specifically, we sweep over a range of $s$, $p$, and $c$ to form the grid, and calculate the $\alpha$ value that binds the chance constraint for each combination. Among the grid, we then select an $s$, $p$, $c$ combination that minimizes (7)'s objective.

The resulting algorithm is Algorithm 1. As discussed above, this algorithm only needs to query MMAA's APGD components once on a small set of validation data, and all other steps are simple mathematical operations requiring minimal computation. Additionally, note that the optimization precision of Algorithm 1 is governed by the discrete nature of the evaluation dataset. I.e., with a dataset consisting of 10,000 examples (such as the CIFAR-10 and CIFAR-100 evaluation sets), the finest optimization accuracy one can expect is 0.01% in terms of objective value (accuracy). Hence, it is not necessary to solve (7) to a high accuracy. Moreover, as shown in Figure 9 in Appendix D.1, which analyzes the sensitivity of the formulation (7)'s objective value with respect to $s$, $p$, and $c$, the optimization landscape is relatively smooth. Therefore, a relatively coarse grid (512 combinations in our case) can find a satisfactory solution, and hence Algorithm 1 is highly efficient despite the triply nested loop structure. Furthermore, the base classifier raw logits associated with $h_{\text{rob}}^{\text{LN}}(\cdot)$'s minimum-margin perturbations do not depend on $s$, $p$, $c$ and can be cached. Hence, the number of forward loops is agnostic to the search space size.

All of the above makes Algorithm 1 efficiently solvable. In practice, the triply-nested grid search loop can be completed within ten seconds on a laptop computer, and performing MMAA on 1000 images requires 3752/10172 seconds for CIFAR-100/ImageNet with a single Nvidia RTX-8000 GPU.

---

[3]To understand this, suppose that for some combination of $s$, $p$, $c$, and $\alpha$, this inequality is satisfied strictly. Then, it will be possible to decrease $\alpha$ (i.e., increase $\frac{1-\alpha}{\alpha}$) without violating this constraint, and thereby further reduce the objective value.

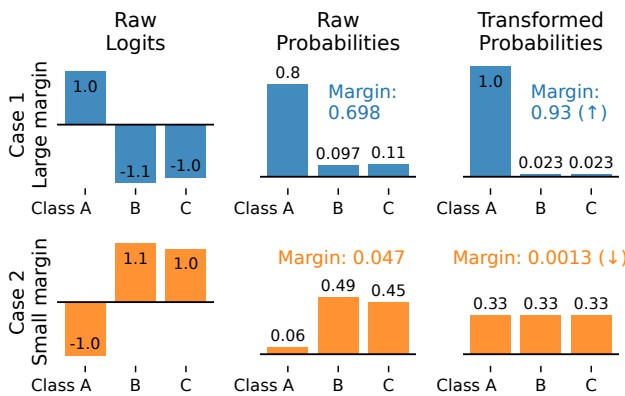
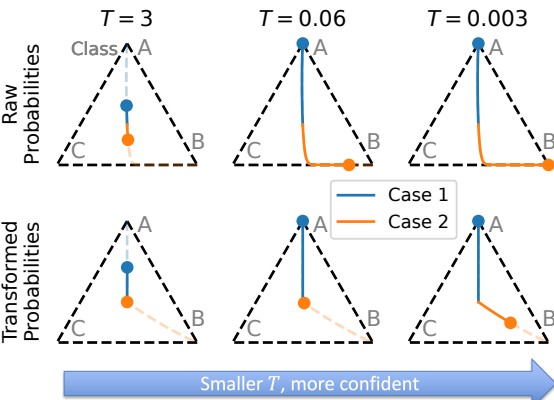

Figure 3: The raw logits, the corresponding prediction probabilities, and the probabilities computed with the transformed logits. Our transformation augments the confidence margin difference between the two scenarios.

Figure 4: Probability trajectories on the probability simplex formed by temperature scaling, with or without the logit transformation. The transformation reduces confidence when classes compete.

### 4.4 Visualization of the Nonlinear Logit Transformation $M(s, p, c)$

To better understand the effects of the proposed nonlinear logit transformation $M(s, p, c)(\cdot)$, we visualize how it affects the base classifier prediction probabilities when coupled with the Softmax operation. Consider a three-class (A, B, and C) classification problem, with two example logit vectors. The first example simulates the case where class A is clearly preferred over the rest (large margin), while the second example illustrates a competition between classes B and C (small margin). The raw logits, the corresponding prediction probabilities, and the probabilities computed with the transformed logits are visualized in Figure 3. Clearly, the margin is further increased for the large margin case and shrunk for the small margin case, which aligns with the goal of enlarging the benign confidence property of the base classifiers.

For further demonstration, we adjust the overall confidence level for the above two cases and compare how their prediction probabilities change with the confidence level. Specifically, by applying temperature scaling and varying the temperature $\tau$, the prediction probability vectors form trajectories on the probability simplex, whose vertices represent the classes.[4] For example, a small temperature $\tau$ increases the overall prediction confidence, moving the vector toward a vertex. Conversely, a large temperature $\tau$ attracts the prediction probability to the simplex's centroid. By continuously adjusting the temperature, we obtain trajectories that connect the centroid to the vertices. By comparing the trajectories formed with or without the nonlinear logit transformation ($\sigma(h_{\mathrm{rob}}(\cdot)/\tau)$ and $\sigma(h_{\mathrm{rob}}^{M(s,p,c)}(\cdot)/\tau)$), we can better understand the transformation's properties.

Figure 4 shows the prediction probability vectors at three example temperature values, as well as the trajectories formed by continuously varying the temperature. We observe that the nonlinear logit transformation significantly slows down the movement of the small margin case from the centroid to the vertex. Moreover, the trajectory with the transformation is straighter and further from the edge BC, implying that the competition between classes B and C has been reduced. In the context of mixed classifiers, the nonlinear transformation reduces the robust base classifier's relative authority in the mixture when it encounters competing classes, thereby improving the mixed classifier's accuracy-robustness trade-off.

## 5 Experiments

We use extensive experiments to demonstrate the accuracy-robustness balance of the MixedNUTS classifier $f_{\mathrm{mix}}^{M(s^\star, p^\star, c^\star)}(\cdot)$, focusing on the effectiveness of the nonlinear logit transformation. Our evaluation uses CIFAR-10 (Krizhevsky, 2009), CIFAR-100 (Krizhevsky, 2009), and ImageNet (Deng et al., 2009) datasets. For

---

[4]Here, the purpose of temperature scaling is different from Section 3.1. In Section 3.1, temperature scaling mitigates $g_{\mathrm{std}}(\cdot)$'s detrimental confidence property. Here, scaling with variable temperatures generates probability trajectories for visualization.

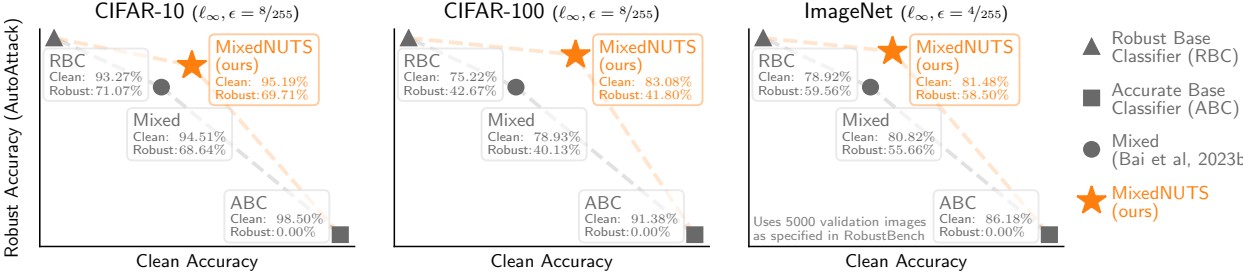

Figure 5: MixedNUTS balances the robustness from its robust base classifier and the accuracy from its standard base classifier. The nonlinear logit transformation helps MixedNUTS achieve a much better accuracy-robust trade-off than a baseline mixed model without transformation. Appendix C.1 reports the base model details and the optimal $s$, $p$, $c$, $\alpha$ values.

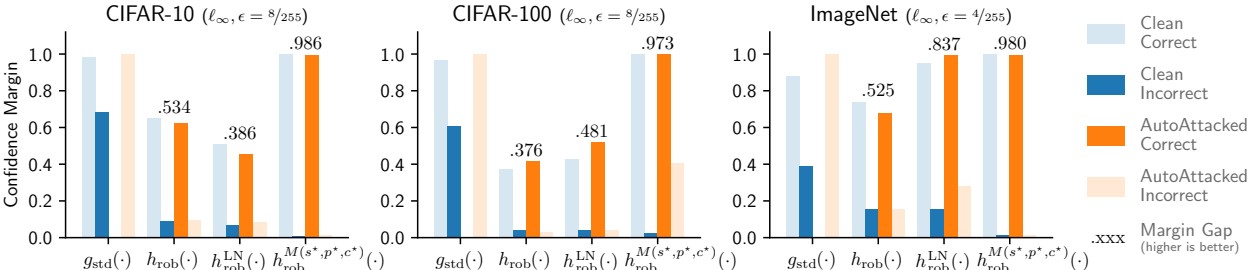

Figure 6: The median confidence margin of the accurate/robust base classifier $g_{\text{std}}(\cdot)/h_{\text{rob}}(\cdot)$, the layer-normed logits $h_{\text{rob}}^{\text{LN}}(\cdot)$, and the nonlinearly transformed model $h_{\text{rob}}^{M(s^\star,p^\star,c^\star)}(\cdot)$ on clean and AutoAttacked data, grouped by prediction correctness. The number above each bar group is the "margin gap", defined as the difference between the medians on clean incorrect inputs and AutoAttacked correct ones. A higher margin gap signals more benign confidence property, and thus better accuracy-robustness trade-off for the mixed classifier.

each dataset, we select the model with the highest robust accuracy verified on RobustBench (Croce et al., 2021) as the robust base classifier $h_{\text{rob}}(\cdot)$, and select a state-of-the-art standard (non-robust) model enhanced with extra training data as the accurate base classifier $g_{\text{std}}(\cdot)$. Detailed model information is reported in Appendix C.1.

Table 1: MixedNUTS's error rate changes relative to the robust base classifier (more negative is better).

| | Clean ($\downarrow$) | Robust (AutoAttack) ($\downarrow$) |
|---|---|---|
| CIFAR-10 | $-28.53\%$ | $+4.70\%$ |
| CIFAR-100 | $-31.72\%$ | $+1.52\%$ |
| ImageNet | $-12.14\%$ | $+2.62\%$ |

As an ensemble method, in addition to being training-free, MixedNUTS is also highly efficient during inference time. Compared with a state-of-the-art robust classifier, MixedNUTS's increase in inference FLOPs is as low as 24.79%. A detailed comparison and discussion on inference efficiency can be found in Appendix C.2.

All mixed classifiers are evaluated with strengthened adaptive AutoAttack algorithms specialized in attacking MixedNUTS and do not manifest gradient obfuscation issues, with the details explained in Appendix B.2.

## 5.1 Main Experiment Results

Figure 5 compares MixedNUTS with its robust base classifier, its accurate base classifier, and the baseline method Mixed (Bai et al., 2024b) on three datasets. Specifically, Mixed is a mixed classifier without the nonlinear logit transformations. Figure 5 shows that MixedNUTS consistently achieves higher clean accuracy and better robustness than this baseline, confirming that the proposed logit transformations mitigate the overall accuracy-robustness trade-off.

Table 1 compares MixedNUTS's relative error rate change over its robust base classifier, showing that MixedNUTS vastly reduces the clean error rate with only a slight robust error rate increase. Specifically, the

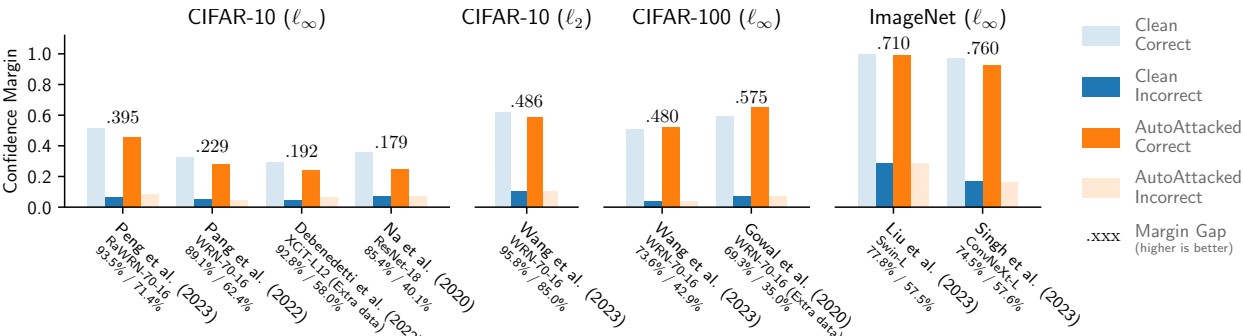

Figure 7: The median confidence margin of a diverse set of robust models with the logits standardized via layer normalization. All models enjoy higher margins on correct predictions than on incorrect ones for clean and adversarial inputs alike. The percentages below each model name are the clean/AutoAttack accuracy.

relative clean error rate improvement is 6 to 21 times more prominent than the relative robust error rate increase. Clearly, MixedNUTS balances accuracy and robustness without additional training.

Figure 6 compares the robust base classifier's confidence margins on clean and attacked data with or without our nonlinear logit transformation (4). For each dataset, the transformation enlarges the margin gap between correct and incorrect predictions, especially in terms of the median which represents the margin majority. Using $h_{\mathrm{rob}}^{M(s^\star, p^\star, c^\star)}(\cdot)$ instead of $h_{\mathrm{rob}}(\cdot)$ makes correct predictions more confident while keeping the mispredictions less confident, making the mixed classifier more accurate without losing robustness.

Figure 2 compares MixedNUTS with existing methods with the highest AutoAttack-validated adversarial robustnesses, confirming that MixedNUTS noticeably improves clean accuracy while maintaining competitive robustness. Moreover, since MixedNUTS can use existing or future improved accurate or even robust models as base classifiers, the entries of Figure 2 should not be regarded as pure competitors.

Existing models suffer from the most pronounced accuracy-robustness trade-off on CIFAR-100, where MixedNUTS offers the most prominent improvement. MixedNUTS boosts the clean accuracy by 7.86 percentage points over the state-of-the-art non-mixing robust model while reducing merely 0.87 points in robust accuracy. In comparison, the previous mixing method (Bai et al., 2024a) sacrifices 3.95 points of robustness (4.5x MixedNUTS's degradation) for a 9.99-point clean accuracy bump using the same base models. Moreover, (Bai et al., 2024a) requires training an additional mixing network component, whereas MixedNUTS is training-free (MixedNUTS is also compatible with the mixing network for even better results). Clearly, MixedNUTS utilizes the robustness of $h_{\mathrm{rob}}(\cdot)$ more effectively and efficiently.

On CIFAR-10 and ImageNet, achieving robustness against common attack budgets penalizes the clean accuracy less severely than on CIFAR-100. Nonetheless, MixedNUTS is still effective in these less suitable cases, reducing the clean error rate by 28.53%/12.14% (relative) while only sacrificing 1.91%/0.98% (relative) robust accuracy on CIFAR-10/ImageNet compared to non-mixing methods. On CIFAR-10, MixedNUTS matches (Bai et al., 2024a)'s clean accuracy while reducing the robust error rate by 5.17% (relative).

With the nonlinear transformation in place, it is still possible to adjust the emphasis between clean and robust accuracy at inference time. This can be achieved by simply re-running Algorithm 1 with a different $\beta$ value. Note that the MMAA step in Algorithm 1 does not depend on $\beta$, and hence can be cached to speed up re-runs. Meanwhile, the computational cost of the rest of Algorithm 1 is marginal. Our experiments use $\beta = 98.5\%$ for CIFAR-10 and -100, and use $\beta = 99.0\%$ for ImageNet. The optimal $s$, $p$, $c$ values and the searching grid used in Algorithm 1 are discussed in Appendix C.1.

## 5.2 Confidence Properties of Various Robust Models

MixedNUTS is built upon the observation that robust models are more confident in correct predictions than incorrect ones. Figure 7 confirms this property across existing models with diverse structures trained with different loss functions across various datasets, and hence MixedNUTS is applicable for a wide range of base

classifier combinations. For a fair confidence margin comparison, all logits are standardized to zero mean and identity variance (corresponding to $h_{\mathrm{rob}}^{\mathrm{LN}}(\cdot)$) before converted into probabilities. Appendix C.3 presents histograms to offer more margin distribution details.

Figure 7 illustrates that existing CIFAR-10 and -100 models have tiny confidence margins for mispredictions and moderate margins for correct predictions, implying that most mispredictions have a close runner-up class. ImageNet classifiers also have higher confidence in correct predictions than incorrect ones. However, while the logits are always standardized before Softmax, the ImageNet models have higher overall margins than the CIFAR ones. This observation indicates that ImageNet models often do not have a strong confounding class despite having more classes, and their non-predicted classes' probabilities spread more evenly.

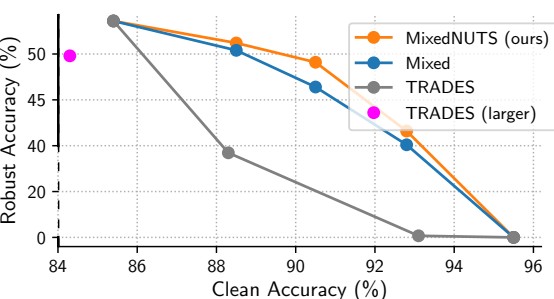

Figure 8: Accuracy-robustness trade-off comparison between MixedNUTS, mixed classifier without nonlinear transformation, and TRADES on 1000 CIFAR-10 images. TRADES (larger) denotes a larger TRADES model trained from scratch that has the same size as MixedNUTS.

### 5.3 Accuracy-Robustness Trade-Off Curves

In Figure 8, we show MixedNUTS's robust accuracy as a function of its clean accuracy. We then compare this accuracy-robustness trade-off curve with that of the mixed classifier without nonlinear logit transformation (Bai et al., 2024b) and that of TRADES (Zhang et al., 2019), a popular adjustable method that aims to improve the trade-off. Note that adjusting between accuracy and robustness with TRADES requires tuning its training loss hyper-parameter $\beta_{\mathrm{TR}}$ and training a new model, whereas the mixed classifiers are training-free and can be adjusted at inference time.

Specifically, we select CIFAR-10 WideResNet-34-10 models trained with $\beta_{\mathrm{TR}} = 0, 0.1, 0.3$, and 6 as the baselines, where 0 corresponds to standard (non-robust) training and 6 is the default which optimizes robustness. For a fair comparison, we use the TRADES models with $\beta_{\mathrm{TR}} = 0$ and 6 to assemble the mixed classifiers. For MixedNUTS, we adjust the level of robustness by tuning $\beta$, the level-of-robustness hyperparameter of Algorithm 1, specifically considering $\beta$ values of 1, 0.96, 0.93, 0.8, and 0.

Figure 8 confirms that training-free mixed classifiers, MixedNUTS and Mixed (Bai et al., 2024b), achieve much more benign accuracy-robustness trade-offs than TRADES, with MixedNUTS attaining the best balance.

Since MixedNUTS is an ensemble, it inevitably results in a larger overall model than the TRADES baseline. To clarify that MixedNUTS's performance gain is not due to the increased size, we train a larger TRADES model (other training settings are unchanged) to match the parameter count, inference FLOPS, and parallelizability. As shown in Figure 8, this larger TRADES model's clean and robust accuracy does not improve over the original, likely because the original training schedule is suboptimal for the increased size. This is unsurprising, as it has been shown that no effective one-size-fits-all adversarial training parameter settings exist (Duesterwald et al., 2019). Hence, an increased inference computation does not guarantee better performance on its own. To make a model benefit from a larger size via training, neural architecture and training setting searches are likely required, which is highly cumbersome and unpredictable. In contrast, MixedNUTS is a training-free plug-and-play add-on, enjoying significantly superior practicality.

## 6 Conclusions

This work proposes MixedNUTS, a versatile training-free method that combines the output probabilities of a robust classifier and an accurate classifier. By introducing nonlinear base model logit transformations, MixedNUTS more effectively exploits the benign confidence property of the robust base classifier, thereby achieving a balance between clean data classification accuracy and adversarial robustness. For performance-driven practitioners, this balance implies less to lose in using robust models, incentivizing the real-world deployment of safe deep learning systems. For researchers, as improving the accuracy-robustness trade-off with

a single model becomes harder, MixedNUTS identifies building base models with better margin properties as a novel alternative direction to improve the trade-off in an ensemble setting.

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

# Appendix

## A  Proofs

### A.1  Proof to Theorem 4.3

**Theorem 4.3 (restated).** *Suppose that Assumption 4.2 holds. Let $r_{f^M_{mix}}$ and $r_{h_{rob}}$ denote the robust accuracy of $f^M_{mix}(\cdot)$ and $h_{rob}(\cdot)$ respectively. If $\beta \geq r_{f^M_{mix}}/r_{h_{rob}}$, then a solution to (3) is feasible for (2).*

*Proof.* Suppose that $M(\cdot)$ is a solution from (3). Since the mixed classifier $f^M_{\mathrm{mix}}(\cdot)$ is by construction guaranteed to be correct and robust at some $x$ if $h^M_{\mathrm{rob}}(\cdot)$ is correct and robust with a margin no smaller than $\frac{1-\alpha}{\alpha}$ at $x$, it holds that

$$\mathbb{P}_{(X,Y)\sim\mathcal{D}}\big[\arg\max_i f^M_{\mathrm{mix},i}(X + \delta^\star_{f^M_{\mathrm{mix}}}(X)) = Y\big] \geq \mathbb{P}_{(X,Y)\sim\mathcal{D}}\big[m_{h^M_{\mathrm{rob},i}}(X + \delta^\star_{f^M_{\mathrm{mix}}}(X)) \geq \tfrac{1-\alpha}{\alpha},\ H^M_{\mathrm{cor}}(X)\big]$$

$$= \mathbb{P}_{(X,Y)\sim\mathcal{D}}\big[m_{h^M_{\mathrm{rob},i}}(X + \delta^\star_{f^M_{\mathrm{mix}}}(X)) \geq \tfrac{1-\alpha}{\alpha}\big|H^M_{\mathrm{cor}}(X)\big] \cdot \mathbb{P}_{(X,Y)\sim\mathcal{D}}[H^M_{\mathrm{cor}}(X)],$$

where $H_{\mathrm{cor}}(X)$ denotes the event of $h_{\mathrm{rob}}(\cdot)$ being correct at $X$, i.e., $\arg\max_i h_{\mathrm{rob},i}(X + \delta^\star_{f^M_{\mathrm{mix}}}(X)) = Y$. Similarly, $H^M_{\mathrm{cor}}(X)$ denotes $\arg\max_i h^M_{\mathrm{rob},i}(X + \delta^\star_{f^M_{\mathrm{mix}}}(X)) = Y$. Under Assumption 4.2, $H_{\mathrm{cor}}(X)$ is equivalent to $H^M_{\mathrm{cor}}(X)$. Therefore,

$$\mathbb{P}_{(X,Y)\sim\mathcal{D}}\big[\arg\max_i f^M_{\mathrm{mix},i}(X + \delta^\star_{f^M_{\mathrm{mix}}}(X)) = Y\big] = \mathbb{P}_{(X,Y)\sim\mathcal{D}}\big[m_{h^M_{\mathrm{rob},i}}(X + \delta^\star_{f^M_{\mathrm{mix}}}(X)) \geq \tfrac{1-\alpha}{\alpha}\big|H_{\mathrm{cor}}(X)\big] \cdot \mathbb{P}_{(X,Y)\sim\mathcal{D}}[H_{\mathrm{cor}}(X)]$$

$$= r_{h_{\mathrm{rob}}} \cdot \mathbb{P}_{X\sim\mathcal{X}'_{\mathrm{adv}}}\big[m_{h^M_{\mathrm{rob},i}}(X + \delta^\star_{f^M_{\mathrm{mix}}}(X)) \geq \tfrac{1-\alpha}{\alpha}\big]$$

$$\geq r_{h_{\mathrm{rob}}} \cdot \mathbb{P}_{Z\sim\mathcal{X}'_{\mathrm{adv}}}\big[m^\star_{h^M_{\mathrm{rob}}}(Z) \geq \tfrac{1-\alpha}{\alpha}\big] \geq r_{h_{\mathrm{rob}}} \cdot \beta \geq r_{f^M_{\mathrm{mix}}},$$

which proves the statement. $\qquad\square$

### A.2  Proof to Theorem 4.4

**Theorem 4.4 (restated).** *Suppose that Assumption 4.1 holds. Furthermore, consider an input random variable $X$ and suppose that the margin of $h^M_{rob}(X)$ is independent of whether $g_{std}(X)$ is correct. Then, minimizing the objective of (3) is equivalent to maximizing the objective of (2).*

*Proof.* By the construction of the mixed classifier, for a clean input $x$ incorrectly classified by $h^M_{\mathrm{rob}}(\cdot)$ (i.e., $x$ is in the support of $\mathcal{X}^{\boldsymbol{\times}}_{\mathrm{clean}}$), the mixed classifier prediction $f_{\mathrm{mix}}(x)$ is correct if and only if $g^{\mathrm{TS}(0)}_{\mathrm{std}}(x)$ is correct and $h^M_{\mathrm{rob}}(x)$'s margin is no greater than $\frac{1-\alpha}{\alpha}$.

Let $G_{\mathrm{cor}}(X)$ denote the event of $g_{\mathrm{std}}(X)$ being correct, i.e., $\arg\max_i g^{\mathrm{TS}(0)}_{\mathrm{std}}(X) = Y$. Furthermore, let $\mathcal{D}_{ic}$ denote the data-label distribution formed by clean examples incorrectly predicted by $h^M_{\mathrm{rob}}(\cdot)$. Then,

$$\mathbb{P}_{(X,Y)\sim\mathcal{D}_{ic}}\big[\arg\max_i f^M_{\mathrm{mix},i}(X) = Y\big] = \mathbb{P}_{X\sim\mathcal{X}^{\boldsymbol{\times}}_{\mathrm{clean}}}\big[m_{h^M_{\mathrm{rob}}}(X) < \tfrac{1-\alpha}{\alpha}, G_{\mathrm{cor}}(X)\big]$$

$$= \mathbb{P}_{X\sim\mathcal{X}^{\boldsymbol{\times}}_{\mathrm{clean}}}\big[m_{h^M_{\mathrm{rob}}}(X) < \tfrac{1-\alpha}{\alpha}\big|G_{\mathrm{cor}}(X)\big] \cdot \mathbb{P}_{X\sim\mathcal{X}^{\boldsymbol{\times}}_{\mathrm{clean}}}[G_{\mathrm{cor}}(X)]$$

for all transformations $M$ and mixing weight $\alpha$ (recall that the mixed classifier $f^M_{\mathrm{mix}}(\cdot)$ depends on $\alpha$).

Suppose that the margin of $h^M_{\mathrm{rob}}(\cdot)$ is independent of the accuracy of $g_{\mathrm{std}}(\cdot)$, then the above probability further equals to

$$\Big(1 - \mathbb{P}_{X\sim\mathcal{X}^{\boldsymbol{\times}}_{\mathrm{clean}}}\big[m_{h^M_{\mathrm{rob}}}(X) \geq \tfrac{1-\alpha}{\alpha}\big]\Big) \cdot \mathbb{P}_{X\sim\mathcal{X}^{\boldsymbol{\times}}_{\mathrm{clean}}}[G_{\mathrm{cor}}(X)]$$

Since $\mathbb{P}_{X \sim \mathcal{X}_{\mathrm{clean}}^{\star}}[G_{\mathrm{cor}}(X)]$ does not depend on $M$ or $\alpha$, it holds that

$$
\underset{M \in \mathcal{M}, \; \alpha \in [1/2, 1]}{\arg\min} \; \mathbb{P}_{X \sim \mathcal{X}_{\mathrm{clean}}^{\star}} \left[ m_{h_{\mathrm{rob}}^M}(X) \geq \tfrac{1-\alpha}{\alpha} \right] = \underset{M \in \mathcal{M}, \; \alpha \in [1/2, 1]}{\arg\max} \; \mathbb{P}_{(X,Y) \sim \mathcal{D}_{ic}} \left[ \arg\max_i f_{\mathrm{mix},i}^M(X) = Y \right]
$$
$$
= \underset{M \in \mathcal{M}, \; \alpha \in [1/2, 1]}{\arg\max} \; \mathbb{P}_{(X,Y) \sim \mathcal{D}} \left[ \arg\max_i f_{\mathrm{mix},i}^M(X) = Y \right],
$$

where the last equality holds because under Assumption 4.1, $h_{\mathrm{rob}}^M(x)$ being correct guarantees $g_{\mathrm{std}}^{\mathrm{TS}(0)}(x)$'s correctness. Since the mixed classifier $f_{\mathrm{mix}}^M(\cdot)$ must be correct given that $h_{\mathrm{rob}}^M(x)$ and $g_{\mathrm{std}}^{\mathrm{TS}(0)}(x)$ are both correct, $f_{\mathrm{mix}}^M(\cdot)$ must be correct at clean examples correctly classified by $h_{\mathrm{rob}}^M(\cdot)$. Hence, maximizing $f_{\mathrm{mix}}^M(\cdot)$'s clean accuracy on $h_{\mathrm{rob}}^M(\cdot)$'s mispredictions is equivalent to maximizing $f_{\mathrm{mix}}^M(\cdot)$'s overall clean accuracy. □

## B  Custom Attack Algorithms

### B.1  Minimum-Margin AutoAttack for Margin Estimation

Our margin-based hyper-parameter selection procedure (Algorithm 1) and confidence margin estimation experiments (Figures 6 and 7) require approximating the minimum-margin perturbations associated with the robust base classifier in order to analyze the mixed classifier behavior. Approximating these perturbations requires a strong attack algorithm. While AutoAttack is often regarded as a strong adversary that reliably evaluates model robustness, its original implementation released with (Croce & Hein, 2020) does not return all perturbed examples. Specifically, traditional AutoAttack does not record the perturbation around some input if it deems the model to be robust at this input (i.e., model prediction does not change). While this is acceptable for estimating robust accuracy, it forbids the calculation of correctly predicted AutoAttacked examples' confidence margins, which are required by Algorithm 1, Figure 6, and Figure 7.

To construct a strong attack algorithm compatible with margin estimation, we propose *minimum-margin AutoAttack (MMAA)*. Specifically, we modify the two APGD components of AutoAttack (untargeted APGD-CE and targeted APGD-DLR) to keep track of the margin at each attack step (the margin history is shared across the two components) and always return the perturbation achieving the smallest margin. The FAB and Square components of AutoAttack are much slower than the two APGD components, and for our base classifiers, FAB and Square rarely succeed in altering the model predictions for images that APGD attacks fail to attack. Therefore, we exclude them for the purpose of margin estimation (but include them for the robustness evaluation of MixedNUTS).

### B.2  Adaptive Attacks for Nonlinearly Mixed Classifier Robustness Evaluation

When proposing a novel adversarially robust model, reliably measuring its robustness with strong adversaries is always a top priority. Hence, in addition to designing MMAA for the goal of margin estimation, we devise two adaptive attack algorithms to evaluate the robustness of the MixedNUTS and its nonlinearly mixed model defense mechanism. Both algorithms are strengthened adaptive versions of AutoAttack. As is the original AutoAttack, both algorithms are ensembles of four attack methods, including a black-box component. Hence, the reported accuracy numbers in this paper are lower bounds to the attacked accuracy associated with each of the components.

#### B.2.1  Transfer-Based Adaptive AutoAttack with Auxiliary Mixed Classifier

Following the guidelines for constructing adaptive attacks (Tramèr et al., 2020), our adversary maintains full access to the end-to-end gradient information of the mixed classifier $f_{\mathrm{mix}}(\cdot)$. Nonetheless, when temperature scaling with $T = 0$ is applied to the accurate base classifier $g_{\mathrm{std}}(\cdot)$ as discussed in Section 3.1, $g_{\mathrm{std}}^{\mathrm{TS}(T)}(\cdot)$ is no longer differentiable. While this is an advantage in practice since the mixed classifier becomes harder to attack, we need to circumvent this obfuscated gradient issue in our evaluations to properly demonstrate white-box robustness. To this end, transfer attack comes to the rescue. We construct an auxiliary differentiable mixed classifier $\widetilde{f}_{\mathrm{mix}}(\cdot)$ by mixing $g_{\mathrm{std}}(\cdot)$'s unmapped logits with $h_{\mathrm{rob}}^M(\cdot)$. We allow our attacks to query the gradient

of $\widetilde{f}_{\mathrm{mix}}(\cdot)$ to guide the gradient-based attack on $f_{\mathrm{mix}}(\cdot)$. Since $g_{\mathrm{std}}(\cdot)$ and $g_{\mathrm{std}}^{\mathrm{TS}(T)}(\cdot)$ always produce the same predictions, the transferability between $\widetilde{f}_{\mathrm{mix}}(\cdot)$ and $f_{\mathrm{mix}}(\cdot)$ should be high.

On the other hand, while $h_{\mathrm{rob}}^{M(s,p,c)}(\cdot)$'s nonlinear logit transformation (4) is differentiable, it may also hinder gradient flow in certain cases, especially when the logits fall into the relatively flat near-zero portion of the clamping function $\mathrm{Clamp}(\cdot)$. Hence, we also provide the raw logits of $h_{\mathrm{rob}}(\cdot)$ to our evaluation adversary for better gradient flow. To keep the adversary aware of the transformation $M(s,p,c)(\cdot)$, we still include it in the gradient (i.e., $M(s,p,c)(\cdot)$ is only partially bypassed). The overall construction of the auxiliary differentiable mixed classifier $\widetilde{f}_{\mathrm{mix}}(\cdot)$ is then

$$\widetilde{f}_{\mathrm{mix}}(x) = \log\left((1-\alpha_d)\cdot\sigma\circ g_{\mathrm{std}}(\cdot) + \alpha_d r_d\cdot\sigma\circ h_{\mathrm{rob}}(\cdot) + \alpha_d(1-r_d)\cdot\sigma\circ h_{\mathrm{rob}}^{M(s^\star,p^\star,c^\star)}(\cdot)\right), \tag{8}$$

where $\alpha_d$ is the mixing weight and $r_d$ adjusts the level of contribution of $M(s,p,c)(\cdot)$ to the gradient. Our experiments fix $r_d$ to 0.9 and calculate $\alpha_d$ using Algorithm 1 with no clamping function, $s$ and $p$ fixed to 1, and $c$ fixed to 0. The gradient-based components (APGD and FAB) of our adaptive AutoAttack use $\nabla L(\widetilde{f}_{\mathrm{mix}}(x))$ as a surrogate for $\nabla L(f_{\mathrm{mix}}^{M(s^\star,p^\star,c^\star)}(x))$ where $L$ is the adversarial loss function. The gradient-free Square attack component remains unchanged. Please refer to our source code for details on implementation.

With the transfer-based gradient query in place, our adaptive AutoAttack does not suffer from gradient obfuscation, a phenomenon that leads to overestimated robustness. Specifically, we observe that the black-box Square component of our adaptive AutoAttack does not change the prediction of any images that white-box components fail to attack, confirming the effectiveness of querying the transfer-based auxiliary differentiable mixed classifier for the gradient. If we set $r_d$ to 0 (i.e., do not bypass $M(s,p,c)(\cdot)$ for gradient), the AutoAttacked accuracy of the CIFAR-100 model reported in Figure 5 becomes 42.97% instead of 41.80%, and the black-box Square attack finds 12 vulnerable images. This comparison confirms that the proposed modifications on AutoAttack strengthen its effectiveness against MixedNUTS and eliminate the gradient flow issue, making it a reliable robustness evaluator.

### B.2.2 Direct Gradient Bypass

An alternative method for circumventing the non-differentiability challenge introduced by our logit transformations is to allow the gradient to bypass the corresponding non-differentiable operations. To achieve so, we again leverage the auxiliary differentiable mixed classifier defined in (8), and construct the overall output as

$$\widetilde{f}_{\mathrm{mix}}(x) + \mathrm{StopGrad}\left(f_{\mathrm{mix}}^{M(s^\star,p^\star,c^\star)}(x) - \widetilde{f}_{\mathrm{mix}}(x)\right),$$

where $\mathrm{StopGrad}$ denotes the straight-through operation that passes the forward activation but stops the gradient (Bengio et al., 2013), for which a PyTorch realization is `Tensor.detach()`. The resulting mixed classifier retains the output values of the MixedNUTS classifier $f_{\mathrm{mix}}^{M(s^\star,p^\star,c^\star)}(x)$ while using the gradient computation graph of the differentiable auxiliary classifier $\widetilde{f}_{\mathrm{mix}}(x)$. In the literature, a similar technique is often used to train neural networks with non-differentiable components, such as VQ-VAEs (Van Den Oord et al., 2017).

This direct gradient bypass method is closely related to the transfer-based adaptive attack described in Appendix B.2.1, but has the following crucial differences:

- **Compatibility with existing attack codebases.** The transfer-based attack relies on the outputs from both $f_{\mathrm{mix}}^{M(s^\star,p^\star,c^\star)}(\cdot)$ and $\widetilde{f}_{\mathrm{mix}}(\cdot)$. Since most existing attack codebases, such as AutoAttack, are implemented assuming that the neural network produces a single output, they need to be modified to accept two predictions. In contrast, direct gradient bypass does not introduce or require multiple network outputs, and is therefore compatible with existing attack frameworks without modifications. Hence, our submission to RobustBench uses the direct gradient bypass method.

- **Calculation of attack loss functions.** From a mathematical perspective, the transfer-based attack uses the auxiliary differentiable mixture to evaluate the attack objective function. In contrast, the direct gradient bypass method uses the original MixedNUTS's output for attack objective calculation, and then uses the gradient computation graph of $\widetilde{f}_{\mathrm{mix}}(\cdot)$ to perform back-propagation. Hence, the resulting gradient is slightly different between the two methods.

Table 2: Details of the base classifiers used in our main experiments.

| Dataset | Robust Base Classifier $g_{\text{std}}(\cdot)$ | Accurate Base Classifier $h_{\text{rob}}(\cdot)$ |
|---|---|---|
| CIFAR-10 | ResNet-152 (Kolesnikov et al., 2020) | RaWideResNet-70-16 (Peng et al., 2023) |
| CIFAR-100 | ResNet-152 (Kolesnikov et al., 2020) | WideResNet-70-16 (Wang et al., 2023) |
| ImageNet | ConvNeXt V2-L (Woo et al., 2023) | Swin-L (Liu et al., 2023) |

Table 3: The optimal $s$, $p$, $c$, $\alpha$ values returned by Algorithm 1 used in our main experiments, presented along with the minimum and maximum candidate values in Algorithm 1's searching grid.

| | $s^\star$ | $c^\star$ | $p^\star$ | $\alpha^\star$ | $s_{\min}$ | $s_{\max}$ | $c_{\min}$ | $c_{\max}$ | $p_{\min}$ | $p_{\max}$ |
|---|---|---|---|---|---|---|---|---|---|---|
| CIFAR-10 | 5.00 | $-1.10$ | 4.00 | .999 | 0.05 | 5 | $-1.1$ | 0 | 1 | 4 |
| CIFAR-100 | .612 | $-2.14$ | 3.57 | .986 | 0.05 | 4 | $-2.5$ | $-0.4$ | 1 | 4 |
| ImageNet | .0235 | $-.286$ | 2.71 | .997 | 0.01 | 0.2 | $-2$ | 0 | 2 | 3 |

Table 4: The proposed nonlinear logit transformation $M(s^\star, p^\star, c^\star)(\cdot)$ has minimal effect on base classifier accuracy.

| Dataset | Clean (full dataset) | | | AutoAttack (1000 images) | | |
|---|---|---|---|---|---|---|
| | $h_{\text{rob}}(\cdot)$ | $h_{\text{rob}}^{\text{LN}}(\cdot)$ | $h_{\text{rob}}^{M(s^\star,p^\star,c^\star)}(\cdot)$ | $h_{\text{rob}}(\cdot)$ | $h_{\text{rob}}^{\text{LN}}(\cdot)$ | $h_{\text{rob}}^{M(s^\star,p^\star,c^\star)}(\cdot)$ |
| CIFAR-10 | 93.27% | 93.27% | 93.25% | 71.4% | 71.4% | 71.4% |
| CIFAR-100 | 75.22% | 75.22% | 75.22% | 43.0% | 42.9% | 43.3% |
| ImageNet | 78.75% | 78.75% | 78.75% | 57.5% | 57.5% | 57.5% |

Our experiments show that when using direct gradient bypass, the original AutoAttack algorithm returns 70.08%, 41.91%, and 58.62% for CIFAR-10, CIFAR-100, and ImageNet respectively with the MixedNUTS model used in Figure 5. Compared with the transfer-based adaptive AutoAttack, which achieves 69.71%, 41.80%, and 58.50%, AutoAttack with direct gradient bypass consistently achieves a lower success rate, but the difference is very small. Hence, we use the transfer-based AutoAttack for Figure 5, but note that both methods can evaluate MixedNUTS reliably.

# C  MixedNUTS Model Details

## C.1  Base Classifier and Mixing Details

Table 2 presents the sources and architectures of the base classifiers selected for our main experiments (Figure 5, Figure 2, Figure 6, and Table 1). The robust base classifiers are the state-of-the-art models listed on RobustBench as of submission, and the accurate base classifiers are popular high-performance models pre-trained on large datasets. Note that since MixedNUTS only queries the predicted classes from $g_{\text{std}}(\cdot)$ and is agnostic of its other details, $g_{\text{std}}(\cdot)$ may be any classifier, including large-scale vision-language models that currently see rapid development.

Table 3 presents the optimal $s^\star$, $p^\star$, $c^\star$, and $\alpha^\star$ values used in MixedNUTS's nonlinear logit transformation returned by Algorithm 1. When optimizing $s$, $p$, and $c$, Algorithm 1 performs a grid search, selecting from a provided set of candidate values. In our experiments, we generate uniform linear intervals as the candidate values for the power coefficient $p$ and the bias coefficient $c$, and use a log-scale interval for the scale coefficient $s$. Each interval has eight numbers, with the minimum and maximum values for the intervals listed in Table 3.

Table 4 shows that MixedNUTS's nonlinear logit transformation $M(s^\star, p^\star, c^\star)(\cdot)$ has negligible effects on base classifier accuracy, confirming that the improved accuracy-robustness balance is rooted in the improved base classifier confidence properties.

Table 5: MixedNUTS's accuracy and inference efficiency compared with state-of-the-art classifiers.

| Model | Architecture | Parameters | GFLOPs | Clean (↑) | AutoAttack (↑) |
|---|---|---|---|---|---|
| CIFAR-10 | | | | | |
| This work | Mixed (see Table 2) | 499.5M | 151.02 | 95.19% | 69.71% |
| Peng et al. (2023) | RaWideResNet-70-16 | 267.2M | 121.02 | 93.27% | 71.07% |
| Bai et al. (2024a) | Mixed with Mixing Net | 566.9M | 117.31 | 95.23% | 68.06% |
| Rebuffi et al. (2021) | WideResNet-70-16 | 266.8M | 77.55 | 92.23% | 66.58% |
| Kolesnikov et al. (2020) | ResNet-152 | 232.3M | 30.00 | 98.50% | 0.00% |
| CIFAR-100 | | | | | |
| This work | Mixed (see Table 2) | 499.5M | 107.56 | 83.08% | 41.80% |
| Wang et al. (2023) | WideResNet-70-16 | 266.9M | 77.56 | 75.22% | 42.67% |
| Bai et al. (2024a) | Mixed with Mixing Net | 567.4M | 117.31 | 85.21% | 38.72% |
| Gowal et al. (2020) | WideResNet-70-16 | 266.9M | 77.55 | 69.15% | 36.88% |
| Kolesnikov et al. (2020) | ResNet-152 | 232.6M | 30.00 | 91.38% | 0.00% |
| ImageNet | | | | | |
| This work | Mixed (see Table 2) | 394.5M | 136.91 | 81.48% | 58.50% |
| Liu et al. (2023) | Swin-L | 198.0M | 68.12 | 78.92% | 59.56% |
| Singh et al. (2023) | ConvNeXt-L + ConvStem | 198.1M | 71.16 | 77.00% | 57.70% |
| Peng et al. (2023) | RaWideResNet-101-2 | 104.1M | 51.14 | 73.44% | 48.94% |
| Woo et al. (2023) | ConvNeXt V2-L | 196.5M | 68.79 | 86.18% | 0.00% |

## C.2 Model Inference Efficiency

In this section, we compare the performance and inference efficiency of MixedNUTS with existing methods. As a training-free ensemble method, MixedNUTS naturally trades inference efficiency for training efficiency. Nonetheless, since MixedNUTS only requires two base models and does not add new neural network components, it is among the most efficient ensemble methods. Specifically, the computational cost of MixedNUTS is the sum of the computation of its two base classifiers, as the mixing operation itself is trivial from a computational standpoint.

In Table 5, the efficiency of MixedNUTS, evaluated in terms of parameter count and floating-point operations (FLOPs), is compared with some other state-of-the-art methods. Compared with the fellow mixed classifier method adaptive smoothing (Bai et al., 2024a), MixedNUTS is more efficient when the base classifiers are the same, as is the case for CIFAR-100. This is because adaptive smoothing introduces an additional mixing network, whereas MixedNUTS only introduces four additional parameters. On CIFAR-10, MixedNUTS uses a denser robust base classifier than adaptive smoothing, with similar number of parameters but higher GFLOPs (121.02 vs 77.56). MixedNUTS's FLOPs count is thus also higher than (Bai et al., 2024a).

## C.3 Base Classifier Confidence Margin Distribution

Table 6 displays the histograms of the confidence margins of the base classifiers used in the CIFAR-100 experiment in Figure 5. We can observe the following conclusions, which support the design of MixedNUTS:

- $g_{\text{std}}(\cdot)$ is more confident when making mistakes under attack than when correctly predicting clean images.
- $h_{\text{rob}}(\cdot)$ is more confident in correct predictions than in incorrect ones, as required by MixedNUTS. Note that even when subject to strong AutoAttack, correct predictions are still more confident than clean unperturbed incorrect predictions.
- Layer normalization increases $h_{\text{rob}}(\cdot)$'s correct prediction margins while maintaining the incorrect margins.
- MixedNUTS's nonlinear logit transformation significantly increases the correct prediction's confidence margins while keeping most of the incorrect margins small.

Table 6: Prediction confidence margin of $h_{\mathrm{rob}}(\cdot)$, $h_{\mathrm{rob}}^{\mathrm{LN}}(\cdot)$, and $h_{\mathrm{rob}}^{M(s^\star,p^\star,c^\star)}(\cdot)$ used in the CIFAR-100 experiments in Figure 6. The nonlinear logit transformation (4) amplifies the margin advantage of correct predictions over incorrect ones. As in Figure 6, 10000 clean examples and 1000 AutoAttack examples are used.

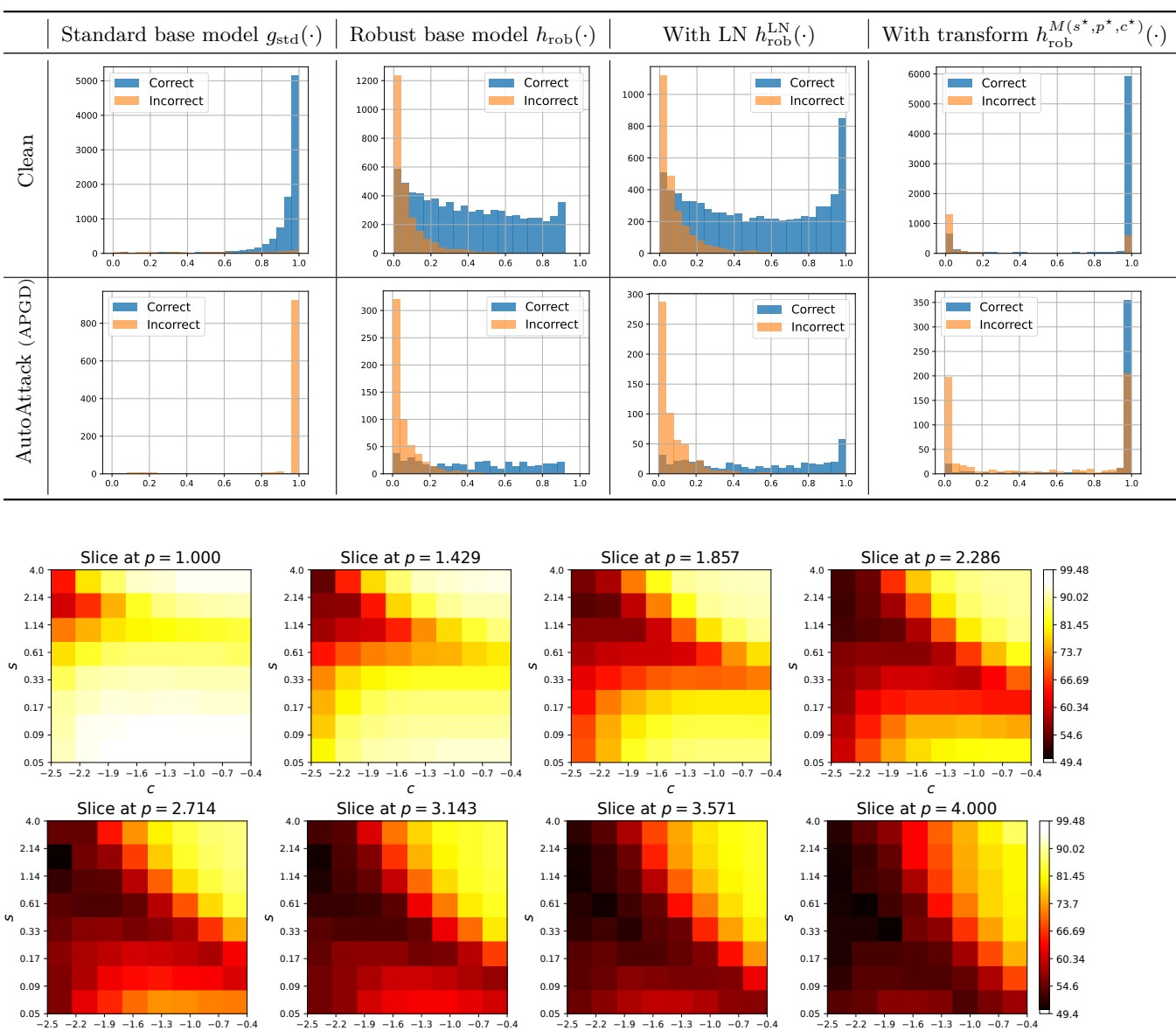

Figure 9: Sensitivity analysis of the nonlinear logit transformation. Lower objective (darker color) is better.

# D  Ablation Studies and Additional Discussions

## D.1  Sensitivity to $s$, $p$, $c$ Values

In this section, we visualize the optimization landscape of the hyper-parameter optimization problem (7) in terms of the sensitivity with respect to $s$, $p$, and $c$. To achieve so, we store the objective of (7) corresponding to each combination of $s$, $p$, $c$ values in our grid search as a three-dimensional tensor (recall that the value of $\alpha$ can be determined via the constraint). We then visualize the tensor by displaying each slice of it as a color map. We use our CIFAR-100 model as an example, and present the result in Figure 9.

Table 7: Ablation study on clamping functions.

| Clamp($\cdot$) | $s^\star$ | $c^\star$ | $p^\star$ | $\alpha^\star$ | $\beta$ | Obj ($\downarrow$) |
|---|---|---|---|---|---|---|
| | | | CIFAR-10 | | | |
| Linear | .050 | - | 9.14 | .963 | .985 | .671 |
| ReLU | 10.4 | $-.750$ | 4.00 | .934 | .985 | .671 |
| GELU | 2.59 | $-.750$ | 4.00 | .997 | .985 | .671 |
| | | | CIFAR-100 | | | |
| Linear | .000005 | - | 10.5 | .880 | .985 | .504 |
| ReLU | .612 | $-2.29$ | 4.00 | .972 | .985 | .500 |
| GELU | .612 | $-2.14$ | 3.57 | .986 | .985 | .500 |

Table 8: The accuracy on images used for calculating $s^\star$, $p^\star$, and $c^\star$ (marked as ✓ in the "Seen" column) is similar to that on images unseen by Algorithm 1 (marked as ✗), confirming the absence of overfitting.

| Dataset | Seen | Clean | AutoAttack |
|---|---|---|---|
| CIFAR-10 | ✓ | 95.20% | 69.20% |
| | ✗ | 95.18% | 69.77% |
| CIFAR-100 | ✓ | 82.80% | 41.60% |
| | ✗ | 83.11% | 41.82% |
| ImageNet | ✓ | 82.60% | 60.80% |
| | ✗ | 81.20% | 57.93% |

Table 9: MixedNUTS's clean and AutoAttack accuracy when $s$, $p$, and $c$ are optimized using different numbers of images. Evaluated with the CIFAR-100 base models from Figure 5 on a 1000-example subset.

| # Images for Optimization | Clean | Auto Attack |
|---|---|---|
| 1000 (Default) | 82.8% | 41.6% |
| 300 | 83.0% | 41.5% |
| 100 | 85.1% | 39.5% |

Table 10: MixedNUTS's clean and AutoAttack accuracy on a 1000-example CIFAR-100 subset with various temperature scales for the standard base model $g_{std}(\cdot)$. The robust base classifier is $h_{rob}^{M(s^\star, p^\star, c^\star)}(\cdot)$ with the $s$, $p$, $c$ values reported in Table 3.

| Accurate Base Model | Clean | Auto Attack |
|---|---|---|
| $g_{std}^{TS(0)}(\cdot)$ (Default) | 82.8% | 41.6% |
| $g_{std}^{TS(0.5)}(\cdot)$ | 82.8% | 41.4% |
| $g_{std}^{TS(1)}(\cdot)$ | 82.8% | 41.3% |

Table 11: The optimal hyperparameters are similar across similar models, and hence can transfer between them. CIFAR-10 models are used, with (Peng et al., 2023) being the model in the main experiments in Figure 5.

| Robust Base Model | $s^\star$ | $c^\star$ | $p^\star$ |
|---|---|---|---|
| (Peng et al., 2023) | 5.0 | $-1.1$ | 4.0 |
| (Pang et al., 2022) | 5.0 | $-1.1$ | 4.0 |
| (Wang et al., 2023) | 5.0 | $-1.1$ | 2.71 |

As shown in Figure 9, while the optimization landscape is non-convex, it is relatively smooth and benign, with multiple combinations achieving similar, relatively low objective values. When the exponent parameter $p$ is small, the other two parameters, $s$ and $c$, have to be within a smaller range for optimal performance. When $p$ is larger, a wide range of values for $s$ and $c$ can work. Nonetheless, an excessively large $p$ may potentially cause numerical instabilities and should be avoided if possible. For the same consideration, we do not recommend using the exponentiation function in the nonlinear logit transformation.

For further illustration, we construct a mixed CIFAR-100 classifier with a simple GELU as the nonlinear logit transformation (still using $g_{std}^{TS(0)}(\cdot)$ as the standard base classifier). The resulting clean/robust accuracy is 77.9%/40.4% on a 1000-example data subset. While this result is slightly better than the 77.6%/39.9% accuracy of the baseline mixed classifier without nonlinearity, it is noticeably worse than MixedNUTS's 82.8%/41.6%. We can thus conclude that selecting a good combination of $s$, $p$, and $c$ is crucial for achieving optimal performance.

## D.2 Selecting the Clamping Function

This section performs an ablation study on the clamping function in the nonlinear logit transformation defined in (4). Specifically, we compare using GELU or ReLU as Clamp($\cdot$) to bypassing Clamp($\cdot$) (i.e., use a linear function). Here, we select a CIFAR-10 ResNet-18 model (Na, 2020) and a CIFAR-100 WideResNet-70-16 model (Wang et al., 2023) as two examples of $h_{rob}(\cdot)$ and compare the optimal objective returned by Algorithm 1 using each of the clamping function settings. As shown in Table 7, while the optimal objective is similar for all three options, the returned hyper-parameters $s^\star$, $p^\star$, $c^\star$, and $\alpha^\star$ is the most "modest" for GELU, which translates to the best numerical stability. In comparison, using a linear clamping function requires applying a power of 9.14 to the logits, whereas using the ReLU clamping function requires scaling

the logits up by a factor of 10.4 for CIFAR-10, potentially resulting in significant numerical instabilities. Therefore, we select GELU as the default clamping function and use it for all other experiments.

### D.3  Effect of Optimization Dataset Size and Absence of Overfitting

Since the MMAA step has the dominant computational time, reducing the number of images used in Algorithm 1 can greatly accelerate it. Analyzing the effect of this data size also helps understand whether optimizing $s$, $p$, and $c$ on validation images introduces overfitting. Table 9 shows that on CIFAR-100, reducing the number of images used in the optimization from 1000 to 300 (3 images per class) has minimal effect on the resulting mixed classifier performance. Further reducing the optimized subset size to 100 still allows for an accuracy-robustness balance, but shifts the balance towards clean accuracy.

To further demonstrate the absence of overfitting, Table 8 reports that under the default setting of optimizing $s$, $p$, $c$ on 1000 images, the accuracy on these 1000 images is similar to that on the rest of the validation images unseen during optimization. The CIFAR-10 and -100 models, in fact, perform slightly better on unseen images. The ImageNet model's accuracy on unseen images is marginally lower than seen ones, likely due to the scarcity of validation images per class (only 5 per class in total since ImageNet has 1000 classes) and the resulting performance variance across the validation set.

### D.4  Temperature Scaling for $g_{\mathrm{std}}(\cdot)$

This section verifies that scaling up the logits of $g_{\mathrm{std}}(\cdot)$ improves the accuracy-robustness trade-off of the mixed classifier. We select the pair of CIFAR-100 base classifiers used in Figure 5. By jointly adjusting the temperature of $g_{\mathrm{std}}(\cdot)$ and the mixing weight $\alpha$, we can keep the clean accuracy of the mixed model to approximately 84 percent and compare the APGD accuracy. In Table 10, we consider two temperature constants: 0.5 and 0. Note that as defined in Section 2.1, when the temperature is zero, the resulting prediction probabilities $\sigma \circ g_{\mathrm{std}}(\cdot)$ is the one-hot vector associated with the predicted class. As demonstrated by the CIFAR-100 example in Table 10, when we fix the clean accuracy to 82.8%, using $T = 0.5$ and $T = 0$ produces higher AutoAttacked accuracy than $T = 1$ (no scaling), with $T = 0$ producing the best accuracy-robustness balance.

### D.5  Algorithm 1 for Black-Box $h_{\mathrm{rob}}(\cdot)$ without Gradient Access

When optimizing the hyper-parameters $s$, $p$, and $c$, Step 2 of Algorithm 1 requires running MMAA on the robust base classifier. While MMAA does not explicitly require access to base model parameters, its gradient-based components query the robust base classifier gradient (the standard base classifier $g_{\mathrm{std}}(\cdot)$ can be a black box).

However, even if the gradient of $h_{\mathrm{rob}}(\cdot)$ is also unavailable, then $s$, $p$, $c$, and $\alpha$ can be selected with one of the following options:

- **Black-box minimum-margin attack.** Existing gradient-free black-box attacks, such as Square (Andriushchenko et al., 2020) and BPDA (Athalye et al., 2018a), can be modified into minimum-margin attack algorithms. As are gradient-based methods, these gradient-free algorithms are iterative, and the only required modification is to record the margin at each iteration to keep track of the minimum margin.
- **Transfer from another model.** Since the robust base classifiers share the property of being more confident in correct examples than in incorrect ones (as shown in Figure 5), an optimal set of $s$, $p$, $c$ values for one model likely also suits another model. So, one may opt to run MMAA on a robust classifier whose gradient is available, and transfer the $s$, $p$, $c$ values back to the black-box model.
- **Educated guess.** Since each component of our parameterization of the nonlinear logit transformation is intuitively motivated, a generic selection of $s$, $p$, $c$ values should also perform better than mixing linearly. In fact, when we devised this project, we used hand-selected $s$, $p$, $c$ values for prototyping and idea verification, and later designed Algorithm 1 for a more principled selection.

To empirically verify the feasibility of transferring hyper-parameters across robust base classifiers, we show that the optimal hyper-parameters are similar across similar models. Consider the CIFAR-10 robust base

classifier used in our main results, which is from (Peng et al., 2023). Suppose that this model is a black box, and the gradient-based components of MMAA cannot be performed. Then, we can seek some similar robust models whose gradients are visible. We use two models, one from (Pang et al., 2022), and the other from (Wang et al., 2023), as examples. As shown in Table 11, the optimal $s$, $p$, $c$ values calculated via Algorithm 1 are highly similar for these three models. Hence, if we have access to the gradients of one of (Peng et al., 2023; Pang et al., 2022; Wang et al., 2023), then we can use Algorithm 1 to select the hyper-parameter combinations for all three models.

Since other parts of MixedNUTS do not require access to base model weights or gradients, MixedNUTS can be applied to a model zoo even when all base classifiers are black boxes.

## D.6 Selecting the Base Classifiers

This section provides guidelines on how to select the accurate and robust base classifiers for the best mixed classifier performance. For the accurate classifier, since MixedNUTS only considers its predicted class and does not depend on its confidence (recall that MixedNUTS uses $g_{\text{std}}^{\text{TS}(0)}(\cdot)$), the classifier with the best-known clean accuracy should be selected. Meanwhile, for the robust base classifier, since MixedNUTS relies on its margin properties, one should select a model that has high robust accuracy as well as benign margin characteristics (i.e., is significantly more confident in correct predictions than incorrect ones). As shown in Figure 7, most high-performance robust models share this benign property, and the correlation between robust accuracy and margins is insignificant. Hence, state-of-the-art robust models are usually safe to use.

That being said, consider the hypothetical scenario that between a pair of robust base classifiers, one has higher robust accuracy and the other has more benign margin properties. Here, one should compare the percentages of data for which the two models are robust with a certain non-zero margin. The model with higher "robust accuracy with margin" should be used.

## D.7 Behavior of Logit Normalization

The LN operation on the model logits makes the margin agnostic to the overall scale of the logits. Consider two example logit vectors in $\mathbb{R}^3$, namely $(0.9, 1, 1.1)$ and $(-2, 1, 1.1)$. The first vector corresponds to the case where the classifier prefers the third class but is relatively unconfident. The second vector reflects the scenario where the classifier is generally more confident, but the second and third classes compete with each other. The LN operation will scale up the first vector and scale down the second. It is likely that the competing scenario is more common when the prediction is incorrect, and therefore the LN operation, which comparatively decreases the margin under the competing scenario, makes incorrect examples less confident compared with correct ones. As a result, the LN operation itself can slightly enlarge the margin difference between incorrect and correct examples.

For ImageNet, instead of performing LN on the logits based on the mean and variance of all 1000 classes, we normalize using the statistics of the top 250 classes. The intuition of doing so is that the predicted probabilities of bottom classes are extremely small and likely have negligible influence on model prediction and robustness. However, they considerably influence the mean and variance statistics of logits. Excluding these least-related classes makes the LN operation less noisy.

## D.8 Further Discussions on Assumptions

### D.8.1 When Assumption 4.1 is Slightly Violated

If Assumption 4.1 is slightly violated, then there is a slight mismatch between the objective functions of (3) and (2) due to discarding the case of $g_{\text{std}}^{\text{TS}(0)}(\cdot)$ being incorrect while $h_{\text{rob}}^{M(s,p,c)}(\cdot)$ being correct on clean data. As a result, the reformulations in this section become slightly suboptimal. However, note that the constraint in (2), which enforces the level of robustness of the mixed classifier, is not compromised. Furthermore, as mentioned above, the amount of clean examples correctly classified by $h_{\text{rob}}^{M(s,p,c)}(\cdot)$ but not by $g_{\text{std}}^{\text{TS}(0)}(\cdot)$ is usually exceedingly rare, and hence the degree of suboptimality is extremely small.

Also note that with a slight violation of Assumption 4.1, while our algorithm may become slightly suboptimal, the mixed classifier outperforms our expectation, because it can now correctly classify additional clean examples than suggested by Theorem 4.4, the only theoretical result dependent on Assumption 4.1.

### D.8.2 When Assumption 4.2 is Slightly Violated

Assumption 4.2 assumes that the nonlinear logit transformation applied to $h_{\mathrm{rob}}(\cdot)$ does not affect its predicted class and hence inherits $h_{\mathrm{rob}}(\cdot)$'s accuracy. When Assumption 4.2 is violated, consider the following two cases: 1) the logit transformation $M(s,p,c)(\cdot)$ corrects mispredictions; 2) $M(s,p,c)(\cdot)$ contaminates correct predictions.

Consider the first scenario, i.e., $h_{\mathrm{rob}}^{M(s,p,c)}(\cdot)$ is correct whereas $h_{\mathrm{rob}}(\cdot)$ is not. In this case, Theorem 4.3 (the only theoretical result dependent on Assumption 4.2) still holds, and the mixed classifier can correctly classify even more clean examples than Theorem 4.3 suggests.

Conversely, consider the second case, where $h_{\mathrm{rob}}^{M(s,p,c)}(\cdot)$ is incorrect whereas $h_{\mathrm{rob}}(\cdot)$ is correct. In this case, Theorem 4.3 may not hold. However, this is the best one can expect. In this worst-case scenario, although the nonlinear logit transformation improves $h_{\mathrm{rob}}(\cdot)$'s confidence property, it also harms $h_{\mathrm{rob}}(\cdot)$'s standalone accuracy, which in turn negatively affects the MixedNUTS model. Fortunately, this worst case is easily avoidable in practice by checking $h_{\mathrm{rob}}^{M(s,p,c)}(\cdot)$'s standalone clean accuracy. If $h_{\mathrm{rob}}^{M(s,p,c)}(\cdot)$'s clean accuracy deteriorates, the search space for $s$, $p$, and $c$ can be adjusted accordingly before re-running Algorithm 1.

### D.8.3 Relaxing the Independence Assumption in Theorem 4.4

Theorem 4.4 assumes that the margin of $h_{\mathrm{rob}}^M(X)$ and the correctness of $g_{\mathrm{std}}(X)$ are independent. Suppose that such an assumption does not hold for a pair of base classifiers. Then, $\mathbb{P}_{X \sim \mathcal{X}_{\mathrm{clean}}^\star}\left[m_{h_{\mathrm{rob}}^M}(X) \geq \frac{1-\alpha}{\alpha}\right]$ may not be equal to $\mathbb{P}_{X \sim \mathcal{X}_{\mathrm{clean}}^\star}\left[m_{h_{\mathrm{rob}}^M}(X) \geq \frac{1-\alpha}{\alpha}\big| G_{\mathrm{cor}}(X)\right]$. In this case, we need to minimize the latter quantity in order to effectively optimize (2). Hence, we need to modify the objective functions of (3) and (6) accordingly, and change the objective value assignment step in Line 10 of Algorithm 1 to $o^{ijk} \leftarrow \mathbb{P}_{X \in \widetilde{\mathcal{X}}_{\mathrm{clean}}^\star}\big[m_{h_{\mathrm{rob}}^{M(s_i,p_j,c_k)}}(X) \geq q_{1-\beta}^{ijk}\big| G_{\mathrm{cor}}(X)\big]$. With such a modification, the optimization of $s$, $p$, $c$ is no longer decoupled from $g_{\mathrm{std}}(\cdot)$, but the resulting algorithm is still efficiently solvable and Theorem 4.4 still holds.

### D.9 Approximation Quality of (7)

Algorithm 1 solves (7) as a surrogate of (6) for efficiency. One of the approximations of (7) is to use the minimum-margin perturbation against $h_{\mathrm{rob}}^{\mathrm{LN}}(\cdot)$ instead of that associated with $h_{\mathrm{rob}}^M(\cdot)$. While $h_{\mathrm{rob}}^M(\cdot)$ and $h_{\mathrm{rob}}^{\mathrm{LN}}(\cdot)$ are expected to have similar standalone accuracy and robustness, their confidence properties are different, and therefore the minimum-margin perturbation associated with $h_{\mathrm{rob}}^M(\cdot)$ can be different from that associated with $h_{\mathrm{rob}}^{\mathrm{LN}}(\cdot)$, inducing a distribution mismatch. To analyze the influence of this mismatch on the effectiveness of Algorithm 1, we record the values of $s^\star$, $p^\star$, and $c^\star$, compute the minimum-margin-AutoAttacked examples of $h_{\mathrm{rob}}^{M(s^\star,p^\star,c^\star)}(\cdot)$ and re-run Algorithm 1 with the new examples. If the objective value calculated via the new examples and $s^\star$, $p^\star$, $c^\star$ is close to the optimal objective returned from the original Algorithm 1, then the mismatch is small and benign and Algorithm 1 is capable of indirectly optimizing (6).

We use the CIFAR-100 model from Figure 5 as an example to perform this analysis. The original optimal objective returned by Algorithm 1 is 50.0%. The re-computed objective based on $h_{\mathrm{rob}}^{M(s^\star,p^\star,c^\star)}(\cdot)$'s minimum-margin perturbations, where $s^\star = .612$, $p^\star = 3.57$, $c^\star = -2.14$, is 66.7%. While there is a gap between the two objective values and therefore the approximation-induced distribution mismatch exists, Algorithm 1 can still effectively decrease the objective value of (6).

