# OpenReview forum: "MixedNUTS: Training-Free Accuracy-Robustness Balance via Nonlinearly Mixed Classifiers"
_TMLR — Accepted by TMLR_

### Review · Reviewer_CcFH · 2024-05-28

**Summary Of Contributions:**

This paper proposes a novel general training-free framework called MixedNUTS, for building a classifier that is both accurate and robust, by combining two general existing models: one with very good accuracy, another with good robustness and reasonable accuracy. The paper proposes nonlinear base model logit transformations on the robust base classifier to exploit its benign confidence property. The paper achieves good numerical results on common image datasets under strong adaptive attacks: MixedNUTS achieved vastly improved accuracy and near-SOTA robustness – it boosts CIFAR-100 clean accuracy by 7.86 points, sacrificing merely 0.87 points in robust accuracy.

**Audience:**

Yes

**Claims And Evidence:**

Yes

**Requested Changes:**

This work would be better and more convincing if (one or two) more attacks are tested in numerical experiments.

**Strengths And Weaknesses:**

Strengths: Overall, this paper has lots of strengths. The most noticeable ones are listed below:
1) The paper provides good intuitive explanations of its various high-level ideas.
2) Under reasonable and almost realistic assumptions, the paper provides rigorous proofs to show its framework is theoretically solid.
3) The paper provides good numerical results achieving best-known accuracy-robustness tradeoff, with significant improvement on this metric over any existing classifier.
4) The method is training-free, and can be applied on one accurate model and one robust model with quite general properties. The handful of parameters can be pre-configured on a small set of data by another algorithm (Algorithm 1), thus don't need much manual tuning.

Weaknesses:
1) Algorithmic novelty: Compared to the existing Mixed method [Bai et al. (2024)], most algorithm change in the proposed MixedNUTS method is through adding a nonlinear logit transformation to the robust classifier. Thus, the algorithm didn't change much, even though the result improved a lot.
2) Most numerical experiments are only tested against two or less attack methods, according to the main paper and appendix B (if I misunderstood, please let me know). The results could be more convincing if more than 3 attacks are included in most experiments.
3) (minor) Theoretically, some assumptions not exactly satisfied in practice. e.g. GELU is not monotonic [assuption 4.2]; clean classifier may not be strictly better than robust classifier on clean data [assumption 4.1]. It would be interesting to touch on the theoretical guarantees when the assumptions are slightly violated, such as when the assumed monotonic activation is replaced by GELU.

---

> ### Author Response · Authors · 2024-07-22
> **Thank you for your suggestions and comments**
>
> Dear Reviewer CcFH,
>
> Thank you for the review and for summarizing the strengths of our methods, including their **realistic assumptions, theoretical rigor, lightweightness, and high performance**, in addition to the **clarity of our paper's explanations**.
>
> We believe that **we have addressed the mentioned questions**, and respond to each question below.
>
> > Q1. Algorithmic novelty: Compared to the existing Mixed method [Bai et al. (2024)], most algorithm changes in the proposed MixedNUTS method are through adding a nonlinear logit transformation to the robust classifier. Thus, the algorithm didn't change much, even though the result improved a lot.
>
> The proposed method significantly improves the accuracy-robustness trade-off by applying one-hot encoding to the accurate base classifier and appending a nonlinear logit transformation to the robust base classifier.
> The **simplicity** of the final form of the proposed method makes it **easy to implement and computationally lightweight, and is therefore an $\textbf{advantage}$** from an application perspective.
>
> **From a theoretical perspective**, while the final method is lightweight to implement, the rigorous analyses on the base classifier confidence margins, the derivation of such a method, and the design of the efficient algorithms are $\textbf{novel, principled, and sophisticated}$.
> To our knowledge, this is the **first work to consider editing confidence properties for improving adversarial robustness** in the heterogeneous ensemble setting.
> On top of this, the reformulations exploit several unique structures of the considered problem, such as the relationship of the two base classifiers (Assumption 4.1) and the transferability between $h^\textrm{LN}$ and $h^M_{s, p, c}$ (Assumption 4.2), to make the originally intractable problem efficiently solvable.
> **Hence, the innovations and contributions of this work are non-trivial.**
>
> Finally, as agreed by the reviewer, the proposed method significantly improves the experimental results. Such an enhancement **highlights the empirical value of our work, in addition to its theoretical novelties.**
>
> $\textbf{In summary, the proposed method is a novel and significant development of the mixed classifier framework.}$
> $\textbf{The high performance and lightweightness of our method are made possible through}$
> $\textbf{delicate analyses and careful reformulations.}$
>
> > Q2. Most numerical experiments are only tested against two or fewer attack methods, according to the main paper and appendix B (if I misunderstood, please let me know). The results could be more convincing if more than three attacks are included in most experiments.
>
> Thank you for the comment. In the adversarial robustness literature, while many types of attack algorithms have been developed, **AutoAttack** [1] **has emerged as one of the most popular evaluation methods** in recent years.
> **Widely regarded as a reliable robustness evaluation benchmark algorithm** [2, 3, 4], AutoAttack is an ensemble of four attack algorithms: APGD-CE (untargeted), APGD-DLR (targeted), FAB (targeted), and Square (untargeted, black-box).
>
> As mentioned in Appendix B.2, our evaluation uses **two enhanced versions of AutoAttack**, specifically strengthened for the structure of MixedNUTS.
>
> Hence, the robust accuracy reported in our paper is **already an** $\textbf{empirical lower bound}$ **to the attacked accuracy evaluated each of AutoAttack's four component algorithms**.
> Furthermore, the gradient-free component of AutoAttack does not find additional adversarial examples on top of gradient-based components, confirming the $\textbf{reliability}$ of the attack used to evaluate MixedNUTS.
>
> [1] Francesco Croce and Matthias Hein. Reliable evaluation of adversarial robustness with an ensemble of diverse parameter-free attacks. \
> [2] Sylvestre-Alvise Rebuffi et al. Fixing data augmentation to improve adversarial robustness. \
> [3] Zekai Wang et al. Better Diffusion Models Further Improve Adversarial Training. \
> [4] Edoardo Debenedetti et al. A light recipe to train robust vision transformers.

---

> ### Author Response · Authors · 2024-07-22
> **Response to Reviewer CcFH (continued)**
>
> > Q3. (minor) Theoretically, some assumptions are not exactly satisfied in practice. e.g. GELU is not monotonic [assuption 4.2]; the clean classifier may not be strictly better than the robust classifier on clean data [assumption 4.1].
> > It would be interesting to touch on the theoretical guarantees when the assumptions are slightly violated, such as when the assumed monotonic activation is replaced by GELU.
>
> ### We first focus on Assumption 4.2.
>
> Assumption 4.2, which assumes the nonlinear logit transformation is monotonic, enforces that **the transformed robust base classifier $h^M_{s, p, c}$ is correct if and only if the raw model $h$ is correct**. When the transformation is not strictly monotonic, it is still quite possible for such a relationship between $h^M_{s, p, c}$ and $h$ to hold, and the subsequent theoretical analyses remain unaffected.
>
> Further relaxing this assumption, **consider the scenario where $h^M_{s, p, c}$ is correct whereas $h$ is not** (i.e., the appended nonlinear transformation corrects mispredictions).
> In this case, Theorem 4.3 (the only theoretical result dependent on Assumption 4.2) still holds, and **the mixed classifier can correctly classify even more clean examples than Theorem 4.3 suggests**.
>
> Conversely, **consider the case where $h^M_{s, p, c}$ is incorrect whereas $h$ is correct**. In this case, Theorem 4.3 may not hold.
> However, **this is the best one can expect**. In this worst-case scenario, although the nonlinear logit transformation improves $h$'s confidence property, it also harms $h$'s standalone accuracy, which in turn negatively affects the MixedNUTS model.
> Fortunately, **this worst case is easily avoidable** in practice by checking $h^M_{s, p, c}$'s standalone clean accuracy. If deterioration is observed, the search space for $s$, $p$, and $c$ can be adjusted accordingly before re-running Algorithm 1.
>
> ### Next, we turn to Assumption 4.1.
>
> Assumption 4.1 assumes that **when a clean example is correctly classified by the robust base classifier $h^M$, it must also be correctly classified by the accurate base classifier $g$**.
> Under this assumption, maximizing the clean accuracy of the mixed classifier $f^M$ is equivalent to decreasing $h^M$'s confident margin when it mispredicts clean examples.
>
> Suppose that Assumption 4.1 is not strictly satisfied, then there is another possibility for $f^M$ to correctly predict clean examples -- $h^M$ correctly predicts whereas $g$ potentially misclassifies. In this scenario, we want to *increase*, instead of decrease, $h^M$'s confidence margin.
>
> For computational efficiency and ease of implementation, our algorithm for optimizing $s$, $p$, and $c$ discards this alternative scenario. Hence, when Assumption 4.1 is slightly violated, there is a slight mismatch between the objective functions of Equations (2) and (3), and the optimization objective used in Algorithm 1 potentially becomes mildly suboptimal. However, since this unaccounted scenario should be rare (if this scenario is common, then a better $g$ should be selected to mix with $h^M$), **the degree of suboptimality should be very small**.
>
> Note that **with a slight violation of Assumption 4.1, while our algorithm may become slightly suboptimal, the mixed classifier $\textbf{outperforms our expectation}$, because it can now correctly classify additional clean examples than suggested by Theorem 4.4**, the only theoretical result dependent on Assumption 4.1.
>
> The above discussions have been added to Appendix D.8 of the revised paper.

---

### Review · Reviewer_C4np · 2024-06-07

**Summary Of Contributions:**

The paper introduces MixedNUTS,  a robust model by ensembling a standard model and a robust model. The motivation stems from the observation that a robust model exhibits greater confidence in correct predictions compared to incorrect ones. Specifically, the ensemble method involves a convex combination of the softmax outputs from the standard model, combined with temperature scaling, and the robust model, combined with a nonlinear transformation.

**Audience:**

Yes

**Claims And Evidence:**

Yes

**Requested Changes:**

Questions I Seek to Have Answered:

1. Why is the nonlinear transformation applied only to the robust base model and not to the non-robust model as well?

2. In Section 4.4, second paragraph, why do we compare $\sigma(h/T)$ with $\sigma(h^M/T)$, considering that the temperature scale $h^M/T$ is not directly used in $f^M$?

3. In Section 5.1, how is "Mixed" chosen? What is $\alpha$ in this combination?

4. Regarding Figure 8, how are the clean accuracy and robust accuracy controlled to produce this plot? Is it by adjusting $\beta$ for MixedNUTS?

Minor Suggestions:

1. I recommend changing the color scheme for the white and grey bars in the bar plots to improve clarity and visual distinction.

**Strengths And Weaknesses:**

Strength:

1. The paper presents a "training-free" method, which is cost-effective once two basic models are available. Since it does not require re-training the entire model, this approach is relatively inexpensive. Moreover, the ensemble method involves a careful combination of the logits/outputs from two trained models, allowing flexibility as the basic models do not need to have the same structure. This makes the method lightweight and empirically easy to use.

2. The paper demonstrates the performance of the proposed method. Experimental results show that MixedNUTS achieves higher clean data accuracy compared to other robust methods while maintaining a reasonable level of robustness.

3. The paper provides extensive discussion and clarity in its writing.

Weaknesses:

1.  A main drawback is the intractability of optimizing parameters for the nonlinear transformation. The paper resorts to an empirical method, grid search, which lack strong theoretical guarantees. For instance, it is challenging to theoretically justify that $h^{LN}$ serves as an effective surrogate for $h^{M}_{s,p,c}$.

---

> ### Author Response · Authors · 2024-07-22
> **Thank you for your suggestions and comments**
>
> Dear Reviewer C4np,
>
> Thank you for the meticulous comments. Furthermore, we appreciate the reviewer for summarizing the strengths of our methods, including **flexibility, cost-effectiveness, lightweightness, and performance, in addition to the clarity of the paper**.
>
> We believe that **we have addressed all mentioned questions**. Below, we respond to each inquiry.
>
> > Q1. A drawback is the intractability of optimizing parameters for the nonlinear transformation. The paper resorts to an empirical method, grid search, which lacks strong theoretical guarantees.
>
> The grid search algorithm is a zeroth-order optimization algorithm suitable for lower-dimensional problems. Since it directly evaluates objective function values, **grid search is** $\textbf{guaranteed}$ **to find an optimal solution** with a large enough search space and a fine enough search grid.
>
> In practice, as shown in Figure 9, the optimization landscape for $s$, $p$, and $c$ is quite smooth.
> Additionally, note that **the optimization precision of Algorithm 1 is governed by the discrete nature of the evaluation dataset**. I.e., with a dataset consisting of 10,000 examples (such as the CIFAR-10 and CIFAR-100 evaluation sets), the finest optimization accuracy one can expect is $0.01\%$ in terms of objective value (accuracy). **Hence, a relatively coarse grid (512 combinations in our case) can find a sufficiently accurate solution.**
>
> As a result, **this grid search algorithm is $\textbf{highly efficient}$, only taking a few seconds** to iterate all $s$, $p$, $c$ combinations.
> If one desires an even more accurate solution (presumably with a larger evaluation set), one can gradually refine the search grid and iteratively perform grid search to balance optimization accuracy and computational complexity.
>
> While grid search can struggle with computational complexity for high-dimensional problems, given the non-convexity, low dimensionality, and moderate optimization accuracy requirement of our considered problem, grid search's robustness and interpretability outweigh its limitations, and is therefore preferred over other algorithms such as gradient search.
>
> $\textbf{In summary, the optimization problem defined in Equation (7) is tractable,}$
> $\textbf{and the grid search algorithm is guaranteed to find optimal solutions in a highly efficient manner.}$
>
> > Q2. For the grid search algorithm, it is challenging to theoretically justify that $h^{\textrm{LN}}$ serves as an effective surrogate for $h^M_{s, p, c}$.
>
> The grid search in Algorithm 1 is built on the minimum-margin perturbations. For optimization efficiency, we use the minimum-margin perturbations associated with $h^{\textrm{LN}}$, which can be pre-calculated, as a surrogate to those associated with $h^M_{s, p, c}$. This substitution leverages the similarity between $h^{\textrm{LN}}$ and $h^M_{s, p, c}$ to avoid exhaustively finding adversarial examples for all $s$, $p$, $c$ combinations.
>
> When Assumption 4.2 is satisfied, i.e., the models $h^M_{s, p, c}$ and $h^{\textrm{LN}}$ have the same predicted classes (it can be satisfied when, e.g., the nonlinear logit transformation $M$ monotonically increases, preserving the ranking of class-wise outputs), if an adversarial perturbation successfully fools $h^{\textrm{LN}}$ into mispredictions, then it also fools $h^M_{s, p, c}$. In other words, **$h^M_{s, p, c}$ and $h^{\textrm{LN}}$ have $\textbf{perfect transferrability}$ and their "adversarial input manifolds" $\textbf{completely overlap}$**.
>
> Hence, if $h^M_{s, p, c}$ achieves high confidence when correctly predicting $h^{\textrm{LN}}$'s adversarial input manifold, it will also be effective on itself's, thereby promoting the robustness of the mixed classifeir. **Therefore, $h^{\textrm{LN}}$ is an effective surrogate for $h^M_{s, p, c}$.*

---

> ### Author Response · Authors · 2024-07-22
> **Response to Reviewer C4np (continued)**
>
> > Q3. Why is the nonlinear transformation applied only to the robust base model and not to the non-robust model as well?
>
> The **robust base classifier $h$ enjoys a** $\textbf{benign confidence property}$ of being more confident in correct predictions than in incorrect ones.
> The intuition for using the nonlinear logit transformation is to **augment** such a crucial property.
> I.e., we want to make $h$'s confident examples (which are likely to be correct) even more confident, so that it can overcome $g$'s mistakes.
> Conversely, we want to further reduce the confidence of $h$'s unconfident predictions (which are likely incorrect), so that it can be corrected by $g$.
>
> **In contrast, the (non-robust) accurate base classifier $g$ does not have this benign confidence property** when subject to adversarial attack.
> In fact, **its confidence property can be the $\textbf{opposite}$ -- more confident in incorrect predictions than correct ones**.
> Hence, we want to **mitigate** this $\textbf{detrimental confidence property}$.
> This divergence in goal between $g$ and $h$ demands different treatments for the two base classifiers.
>
> Note that while mitigating $g$'s detrimental confidence property, we also want to keep its high clean accuracy, i.e., preserve the ranking of each class' prediction $g_i$.
> Given these two objectives, **the best we can expect is enforcing a constant confidence margin** for the accurate base classifier, which is achieved by applying temperature scaling with $T=0$: **the confidence margin of the resulting one-hot encoding $g^{\textrm{TS}, 0}$ is precisely $1$ everywhere**.
>
> $\textbf{In summary, the nonlinear logit transformation is applied to the robust base classifier $h$ to augment its benign}$
> $\textbf{confidence property, but NOT to the accurate base classifier $g$ because its confidence property is NOT benign.}$
> $\textbf{Instead, we use one-hot encoding to enforce a constant margin, eliminating $g$'s detrimental confidence property.}$
>
> We have added the above discussions to Sections 3 and 4 of the revised paper.
>
> > Q4. In Section 4.4, second paragraph, why do we compare $\sigma (h/T)$ with $\sigma (h^M/T)$, considering that the temperature scale $h^M/T$ is not directly used in $f^M$?
>
> Thank you for the question. We believe that **this question is based on $\textbf{misunderstanding}$**. We agree that the explanations in our original submission were a little misleading. We apologize for the confusion and would like to address it here.
>
> The reviewer is correct that **temperature scaling is not applied to the robust base classifier $h$ in the mixing formulation**. Instead, our nonlinear logit transformation $M_{s, p, c}$ can be regarded as a generalization of temperature scaling, with vanilla temperature scaling's functionalities included in the parameter $s$.
>
> **The "temperature scaling" in Section 4.4 is not a part of the mixing formulation, and serves a different purpose from the scaling in Section 3.1.**
> In Section 3.1, temperature scaling mitigates $g$'s detrimental confidence property.
> **In Section 4.4, scaling with various temperatures generates prediction probability trajectories in the probability simplex** $\textbf{for visualization purposes only}$.
> To reduce confusion, we have changed the notation of temperature in Section 4.4 to $\tau$ to differentiate it from the $T$ notation used in Section 3.1.
>
> Specifically, in Section 4.4, we hope to visualize the effect of the nonlinear logit transformation.
> To this end, by applying temperature scaling and varying the temperature $\tau$, the prediction probability vectors form trajectories on the probability simplex, whose vertices represent the classes.
> For example, a small temperature $\tau$ increases the overall prediction confidence, moving the vector toward a vertex.
> Conversely, a large temperature $\tau$ attracts the prediction probability to the simplex's centroid.
> By continuously adjusting the temperature between $+\infty$ and $0$, we obtain trajectories that connect the centroid to the vertices.
> **By comparing the trajectories formed with or without the nonlinear logit transformation ($\sigma (h/\tau)$ and $\sigma (h^M / \tau)$), we can better understand the transformation's properties.**
> The comparison results are explained in the last paragraph of Section 4.4.
>
> We have added the above explanation to Section 4.4 of the revised paper.

---

> ### Author Response · Authors · 2024-07-22
> **Response to Reviewer C4np (continued 2)**
>
> > Q5. In Section 5.1, how is "Mixed" chosen? What is $\alpha$ in this combination?
>
> **"Mixed" denotes the baseline mixed classifier without the proposed nonlinear logit transformations**, as proposed in [1].
> Here, **"Mixed" and MixedNUTS use the same base classifiers** for a fair comparison.
>
> For "Mixed" without the nonlinear logit transformations, the **$\alpha$ values can be selected by running Algorithm 1 with the search space for $s$, $p$, $c$ reduced to a singleton** ($s = 1$, $p = 1$, and $c = 0$).
> In our experiments, $\alpha$ is $0.957$ for CIFAR-10, $0.970$ for CIFAR-100, and $0.907$ for ImageNet.
>
> As later confirmed in Figure 8, the better accuracy-robustness balance of MixedNUTS can be observed across the trade-off curve, and is not restricted to a particular level of robustness.
>
> > Q6. Regarding Figure 8, how are the clean accuracy and robust accuracy controlled to produce this plot? Is it by adjusting $\beta$ for MixedNUTS?
>
> The clean-robust accuracy trade-off curve of MixedNUTS, presented in Figure 8, is **indeed obtained by tuning $\beta$**, the level-of-robustness hyperparameter of Algorithm 1.
> The considered $\beta$ values are $1$, $0.96$, $0.93$, $0.8$, and $0$. We have added this information to the paper.
>
> > Q7. I recommend changing the color scheme for the white and grey bars in the bar plots to improve clarity and visual distinction.
>
> Thank you for the suggestion. We have changed the colors so that now blue bars represent clean data and orange bars represent adversarial data.
>
> The difference between "clean incorrect" and "AutoAttacked correct" confidence margins quantifies the "benign confidence property" that the mixed classifier relies on, and hence the corresponding bars are emphasized over other bars.
>
> [1] Yatong Bai, Brendon G Anderson, and Somayeh Sojoudi. Mixing Classifiers to Alleviate the Accuracy-Robustness Trade-Off.

---

### Review · Reviewer_oyU1 · 2024-07-13

**Summary Of Contributions:**

The authors of the paper propose a novel method to improve the adversarial robustness of a neural classifier, while not losing out on it's prediction accuracy. Their  model is an heterogeneous ensemble of 2 classifiers
- a base classifier ($g(\cdot)$) trained for the attack-free scenario
- robust classifier ($h(\cdot)$) trained to be robust against adversarial attacks

And the ensemble is given by:

$$f_i(x) := log ((1 −\alpha) \cdot \sigma \circ g(x)_i + \alpha \cdot \sigma \circ h(x)_i$$

where $\alpha \in [1/2,1]$ adjusts the mixing weight (given my a previous paper) and $i \in [C]$ for $C$ classes. The base classifier's ($g(\cdot)$) is scaled using temperature scaling

$$g_i^{TS,T} := \frac{g_i(x}{T}$$

and $T$ is set to 0 i.e. the base classifier becomes $g_i^{TS,0}$. To simplify the optimization, the authors make a few assumptions regarding the nature of $h^M(\cdot)$ , where $M$ is the non-linear transform applied on the logits of $h(\cdot)$. The transformed $h(\cdot)$ is defined to be $h^{M}\substack{s\\p\\c}(x)$ where $s \in (0, +\infty)$ is a scaling constant, $p \in (0, +\infty)$ is the exponent constant and $c \in \mathbb{R}$ is a bias constant with the choice of $Clamp$ function being the GELU.  The authors then define an algorithm to optimize for $s, p, c$ and $\alpha$ while also providing a method to calculate a minimum margin for the optimization using the AutoAttack algorithm. The authors also provide extensive empirical studies to back up their claim of an efficient and robust Mixed model.

**Audience:**

Yes

**Broader Impact Concerns:**

There are no broader ethical impacts from this paper.

**Claims And Evidence:**

Yes

**Requested Changes:**

- (Minor) In Section 3: Is is stated that the robust classifier $h(\cdot)$ enjoys the property of having more confidence in correct predictions than in mistakes ... this sentence is a little unclear with the definitions and could use an example or definition in context.
- (Minor) In various sections, the base and robust classifier are defined by $g(\cdot)$ and $h(\cdot)$ respectively. Changing the variable names to something more semantic might help with the readability and referencing of the models in later sections. I would always get confused as to which one was the robust and which one was the base.
- (Minor) In section 4 , when defining the distributions $X_{ic}$ for the distribution formed by clean examples incorrectly classified by $h^M(\cdot)$ and  $X_{ca}$ for the distribution formed by attacked examples correctly classified by $h^M(\cdot)$, it would be a bit more clear to use more descriptive subscripts for the, just to make it easier to refer to later.

**Strengths And Weaknesses:**

The authors do a very good job of explaining the motivation of why this problem is an important one. The paper has a good logical flow, with the authors introducing the problem and giving a high level flowchart of the proposed solution in Figure 1. The authors have also done a comprehensive job is mentioning related papers and prior art. In section 2, the authors do great job of motivating the intuition of using a heterogeneous ensemble classifier is clear, with a clear explanation on the benefits of mixing the outputs of the 2 types of classifiers i.e. the base and the robust. Section 4 is also well written, with the assumptions for the robust classifier clearly stated. The assumptions for the bounds of $\alpha$ are based on a cited paper, when seems reasonable. The definition of the non-linear transform $M$ is also well written, with some issues with the notation here and there. The empirical results and experiments all look reasonable to me.

#### Questions/Clarifications
- In section 3, the authors introduce an explanation for the modifications to the base classifier using temperature scaling. It is a little unclear on why the temperature for the scaled base classifier ($g_i^{TS,T}$)is set to 0. The motivation given for this is  "... to mitigate $g(\cdot)$'s detrimental property of being more confident when mispredicting under attack than when correctly predicting clean data ... ", but it's still unclear why one-hot encoding the outputs would help in this scenario, apart from making the math in section 4 easier.
- In section 4.3, the algorithm for finding $s, p, c$ and $\alpha$ and with the MMAA algorithm maintaining the minimum margin, it seems like the algorithm would be reasonably fast to run as a grid search. It was not clear to me from the section if the search for the hyper parameters would be reasonably efficient based on the size of the search space.
- Also in section 4.3, the search algorithm depends on generating clean examples that are incorrectly classified $\tilde{\chi}_{ic}$ and attacked examples correctly classified $\tilde{\chi}_{ca}$ , both for $h^{LN}(\cdot)$. Would this lead to a cold start problem for the ensemble classifier?

---

> ### Author Response · Authors · 2024-07-22
> **Thank you for your suggestions and comments**
>
> Dear Reviewer oyU1,
>
> Thank you for sharing these constructive comments. We also appreciate the reviewer for acknowledging our motivation, explanations, as well as the theoretical and experimental results.
>
> **We believe that we have addressed the reviewer's questions**. Below, we respond to each inquiry.
>
> > Q1. In section 3, the authors introduce an explanation for the modifications to the base classifier using temperature scaling.
> > It is a little unclear on why the temperature $T$ for the scaled base classifier $g^{\textrm{TS}, T}$ is set to 0. \
> > The motivation given for this is to mitigate $g$'s detrimental property of being more confident when mispredicting under attack than when correctly predicting clean data, but it's still unclear why one-hot encoding the outputs would help in this scenario, apart from making the math in Section 4 easier.
>
> Thank you for the comment. We first clarify the accurate base classifier **$g$'s detrimental confidence property**, and then explain how the proposed one-hot encoding mitigates its damages.
>
> We observe that $g$ is, unfortunately, more confident when mispredicting under attack than when correctly predicting clean data.
> This is detrimental to the mixed classifier, because it is hard to take advantage of $g$ when it makes correct predictions (due to low confidence), and also hard to overcome $g$'s incorrect predictions (due to high confidence).
> Therefore, the mixed classifier's performance is negatively affected by $g$'s confidence property on both clean and adversarial data.
>
> By using $g^{\textrm{TS}, 0}$, which scales with a temperature of zero and is equivalent to **one-hot encoding, we make sure that the confidence margin is $\textbf{equal}$ for correct and incorrect predictions** (the margin is now always $1$), **$\textbf{eliminating}$ $g$'s detrimental confidence property.**
>
> Note that we still hope to preserve the ranking of class-wise outputs $g_i (\cdot)$, so that we can preserve the high accuracy of $g (\cdot)$.
> Given this requirement, **eliminating $g$'s detrimental confidence property by enforcing consistent margin is the best one can expect**.
>
> The above discussions have been added to Section 3.1 of the revised paper.
>
> > Q2. In section 4.3, with the MMAA algorithm finding the minimum-margin adversarial perturbation in advance, the grid search for finding $s$, $p$, $c$, and $\alpha$ seems reasonably fast.
> > It was not clear to me if the search for the hyperparameters would be reasonably efficient based on the size of the search space.
>
> Note that while there are four explicit variables to optimize ($s$, $p$, $c$, $\alpha$), the grid search only applies to $s$, $p$, and $c$, whereas $\alpha$ is calculated in closed form from the constraints of Equation (7) (details are in Algorithm 1). Hence, **the grid search dimension is only three**.
>
> During the grid search, we select eight candidate values for each variable. Hence, there are **only 512 total combinations to compare**.
> Since the minimum-margin perturbations and the associated raw logits from the base classifiers can be pre-computed, **the number of model forward passes is agnostic to the search space size**. I.e., comparing 512 combinations does not involve 512 model passes and is thus $\textbf{very fast}$.
> In practice, the grid search can be completed **within two seconds on a laptop computer**.
>
> As shown in Figure 9, the optimization landscape for this grid search is relatively smooth. Additionally, note that **the optimization precision of Algorithm 1 is governed by the discrete nature of the evaluation dataset**. I.e., with a dataset consisting of 10,000 examples (such as the CIFAR-10 and CIFAR-100 evaluation sets), the finest optimization accuracy one can expect is $0.01\%$ in terms of objective value (accuracy). **Hence, a relatively coarse grid (512 combinations in our case) can find a sufficiently accurate solution.**
>
> If one desires an even more accurate solution (presumably with a larger evaluation set), one can gradually refine the search grid and iteratively perform grid search to balance optimization accuracy and computational complexity.
>
> While grid search can struggle with computational complexity for high-dimensional problems, given the non-convexity, low dimensionality, and moderate optimization accuracy requirement of our considered problem, grid search's robustness and interpretability outweigh its limitations, and is therefore preferred over other algorithms such as gradient search.
>
> **To summarize, the grid search space for $s$, $p$, and $c$ need not be large, the algorithm is highly efficient, and the search can be completed in a few seconds.** We have added the above discussions to Section 4.3 of the revised paper.

---

> ### Author Response · Authors · 2024-07-22
> **Response to Reviewer oyU1 (continued)**
>
> > Q3. Also in section 4.3, the search algorithm depends on generating clean examples that are incorrectly classified ($\\tilde{\\mathcal{X}}\_{ic}$) and attacked examples' correctly classified ($\\tilde{\\mathcal{A}}\_{ca}$), both for $h^{\\textrm{LN}}$.
> > Would this lead to a cold start problem for the ensemble classifier?
>
> In our mixing formulation, the accurate base classifier $g$ and the robust base classifier $h$ are both **already-trained and frozen**, with the transformed model $h^M_{s, p, c}$ only incorporating three additional parameters. Hence, **$h^M_{s, p, c}$ and $h$'s predictions are highly similar**, except the confidence margin property is made more benign. **Hence, the "cold start" problem does not apply to our proposed method.**
>
> In fact, when the nonlinear logit transformation $M$ preserves the ranking of the class-wise outputs of $h$ (one way to satisfy this is to use a non-decreasing "clamping function" in $M$, as suggested in Assumption 4.2), the predicted classes of $h$, $h_{\textrm{LN}}$, and $h^M_{s, p, c}$ are the same. Therefore, the **sets of adversarial examples corresponding to $h$, $h_{\textrm{LN}}$, and $h^M_{s, p, c}$ perfectly coincide**.
>
> **Hence, using $h^{\\textrm{LN}}$ to obtain the sets $\\tilde{\\mathcal{X}}\_{ic}$ and $\\tilde{\\mathcal{A}}\_{ca}$ for optimizing $s$, $p$, and $c$ is highly effective.**
>
> > Q4. (Minor) In Section 3: Is is stated that the robust classifier enjoys the property of having more confidence in correct predictions than in mistakes.
> > This sentence is a little unclear with the definitions and could use an example or definition in context.
>
> Thank you for the suggestion. A more rigorous version of this statement based on the definition of *confidence margin* is "***$h (\cdot)$'s confidence margin is much higher when it makes correct predictions***." Intuitively, even if some input is subject to attack (which vastly decreases the confidence margin of correct predictions), if it is correctly predicted, its margin is still expected to be larger than incorrectly predicted natural examples.
>
> As a result, when mixing the output probabilities $\sigma \circ h$ and $\sigma \circ g$ on clean data, where $g$ is expected to be more accurate than $h$, the model $g$ can correct $h$'s mistake because $h$ is unconfident.
> Meanwhile, when the mixed classifier is under attack and $h$ becomes much more reliable than $g$, model $h$'s high confidence in correct predictions can overcome $g$'s misguided outputs.
> Hence, even when $g$'s robust accuracy is near zero, the mixed classifier still inherits most of $h$'s robustness.
> **Combining the above two cases, we can see that the "benign confidence property" of $h$ allows the mixed classifier to simultaneously take advantage of $g$'s high clean accuracy and $h$'s adversarial robustness.**
> As a result, modifying and enhancing the base classifiers' confidence has vast potential to further improve the mixed classifier.
>
> At the beginning of Section 3 of the revised paper, we have expanded the sentence to provide more rigorous clarifications and intuitive explanations.
>
> > Q5. (Minor) In various sections, the base and robust classifier are defined by $g$ and $h$ respectively. Changing the variable names to something more semantic might help with the readability and referencing of the models in later sections.
>
> > Q6. In section 4, when defining the distributions $\\mathcal{X}\_{ic}$ for the distribution formed by clean examples incorrectly classified by $h^M$ and $\\mathcal{X}\_{ca}$ for the distribution formed by attacked examples correctly classified by, it would be a bit clearer to use more descriptive subscripts.
>
> Thank you for the suggestions. We agree that the notations for the accurate and robust base classifiers as well as the corresponding data subsets can be made more semantically intuitive. However, we believe that changing the notation throughout the paper at this stage of the discussion process will introduce further confusion to the reviewing process. Hence, **we suggest postponing the potential notational improvements until most discussions have concluded**.

---

### Author Response · Authors · 2024-07-22
**Message from the Authors**

We would like to thank all reviewers for carefully reviewing our submission and offering constructive comments and suggestions.
We regard this discussion as a valuable opportunity to improve this paper.

**We also feel encouraged to see that the reviewers agree that the proposed method combines lightweightness, ease of implementation, and high performance, and that our paper clearly expresses the contributions.**

Meanwhile, we believe that **all questions raised by the reviewers can be sufficiently addressed**.
We have provided a detailed response to each comment and revised the paper accordingly, with the main modifications highlighted in blue.
If there are further questions or confusion, please feel free to let us know!

---

### Decision · Action_Editor_1uRC · 2024-08-13

**Recommendation:** Accept as is

**Comment:**

This paper proposed a simple yet effective method to improve adversarial robustness, by combining the logits of robust and non-robust classifiers in the inference stage. The initial submission had several weaknesses, as pointed out by the reviewers. The authors did a successful rebuttal and revision, which led to a unanimous consensus on accepting this submission from all reviewers.

**Audience:**

Of broad interest to adversarial machine learning

**Claims And Evidence:**

Yes. This paper provides empirical evidence to show the proposed method improves adversarial robustness.